# Prompt Curriculum Learning for Efficient LLM Post-Training

**Zhaolin Gao**[1,2], **Joongwon Kim**[1,3], **Wen Sun**[2], **Thorsten Joachims**[2], **Sid Wang**[1],
**Richard Yuanzhe Pang**[1], **Liang Tan**[1]

[1] Meta Superintelligence Labs, [2] Cornell University, [3] University of Washington

## Abstract

Reinforcement learning (RL) is widely used to post-train large language models for tasks such as mathematical reasoning and coding. However, the convergence of RL training remains sensitive to batching and prompt selection strategies. We investigate the factors that affect convergence, including batch size and prompt difficulty. Through large-scale experiments across multiple models and datasets, we show that there exists an optimal batch size that balances generation time and gradient quality, and that prompts of intermediate difficulty (where the model has roughly a 50% chance of success) are the most sample-efficient for model convergence. Motivated by these findings, we propose **Prompt Curriculum Learning** (PCL), a lightweight algorithm that selects intermediate-difficulty prompts using a learned value model. PCL avoids costly rollouts and efficiently guides training by focusing on the most informative samples. Empirically, PCL either achieves the highest performance or requires significantly less training time to reach comparable performance across a suite of benchmarks. Compared to using rollouts to filter, PCL is 12.1× and 16.9× faster on identifying intermediate-difficulty prompts when training on MATH and DeepScaleR respectively.

## 1 Introduction

Recent large language models (LLMs), such as OpenAI-o1 (OpenAI, 2024b) and DeepSeek-R1 (DeepSeek-AI, 2025), have demonstrated strong performance by producing long chain-of-thought (CoT) solutions (Wei et al., 2023; DeepSeek-AI, 2025; Zeng et al., 2025). A key driver of these improvements is reinforcement learning (RL) with rule-based rewards, using algorithms such as PPO (Schulman et al., 2017) and GRPO (Shao et al., 2024). By generating responses online from the current model, RL enables LLMs to self-explore and iteratively improve based on their own outputs.

Substantial effort has been devoted to improving both the performance and efficiency of RL training for LLMs (Brantley et al., 2025; Xu et al., 2025; An et al., 2025; Sun et al., 2025). A recurring insight across recent works (Yu et al., 2025; Zhang et al., 2025; Zheng et al., 2025) is that training on prompts of intermediate difficulty (i.e., neither too easy nor too hard for the current policy) yields significantly better data efficiency. However, existing approaches on identifying intermediate prompts typically rely on either rollouts from the current model or a dictionary that tracks average rewards from previous epochs. The former introduces substantial training overhead due to the high cost of online generation, while the latter suffers from off-policyness especially when the dataset is large. In addition, while these works primarily focus on prompt difficulty, many hyperparameters (e.g., batch size) can significantly affect convergence but remain underexplored in prior work.

In this paper, we systematically study how **batch configuration** and **prompt selection** jointly affect the convergence of RL training, and we use these insights to design a new, compute-efficient curriculum algorithm. We uncover two key findings. First, **there exists an optimal batch size that achieves the best trade-off between faster generation time and smaller gradient noise.** While larger batches reduce gradient noise and allow for higher learning rates, they also increase generation time, limiting update frequency. We identify a sweet spot at the transition point between sublinear

---

zg292@cornell.edu

Figure 1: We conduct a systematic investigation of the trade-offs on generation time vs. batch size and number of prompts vs. generations per prompt. We identify an optimal batch size that achieves the best trade-off and discover that the prompts of intermediate difficulty are the most effective for learning. Building on these insights, we introduce **Prompt Curriculum Learning** (PCL), which trains a value model online for prompt filtering. Compared to the rollout-based filter method, PCL is 12.1× and 16.9× faster during prompt filtering when training on MATH and DeepScaleR respectively.

and linear generation time growth, where convergence speed is maximized. Second, **prompts of intermediate difficulty are the most effective for learning.** When a prompt is too easy or hard, gradient signals tend to vanish, leading to wasted compute. In contrast, prompts for which the model has a ~50% success rate require fewer samples to obtain informative updates. We validate this finding empirically across models, datasets, and batch configurations.

Building on these insights, we introduce **Prompt Curriculum Learning** (PCL), an efficient algorithm that dynamically selects prompts of intermediate difficulty using a value model. At each step, PCL samples a large pool of candidate prompts, predicts their expected reward with a single forward pass, and greedily selects those closest to a target threshold (e.g., 0.5). This approach avoids the overhead of rollout-based prompt filtering while also being much more on-policy than dictionary-based methods. We benchmark PCL across a wide range of models and datasets, including Qwen3-Base (1.7B, 4B, 8B) and Llama3.2-it (3B) on MATH, Olympiad-Bench, Minerva MATH, AMC, and AIME. Empirically, PCL either achieves the highest performance or requires substantially less training time to reach comparable performance.

## 2 PROBLEM SETUP

Let $x$ denote a prompt (e.g., a math question), and let $y$ denote a sampled solution of length $|y|$ generated autoregressively from a policy $\pi$, i.e., $y \sim \pi(\cdot \mid x)$. We assume a binary reward function $r(x,y) \in \{0,1\}$, where $r(x,y) = 1$ if the final answer in $y$ is correct and 0 otherwise. Since the reward is binary, we denote $p_\pi(x) \coloneqq \mathbb{E}_{y \sim \pi(\cdot \mid x)}[r(x,y)]$ as the probability of generating a correct answer from policy $\pi$ on prompt $x$, and $A(x,y) \coloneqq r(x,y) - p_\pi(x)$ as the advantage. To optimize $\pi$, we adopt the purely on-policy variant of GRPO (Shao et al., 2024; DeepSeek-AI, 2025), without KL regularization to a fixed reference policy $\pi_{\text{ref}}$ (Yu et al., 2025) and without standard deviation-based advantage regularization (Liu et al., 2025), by maximizing:

$$\mathbb{E}_{x \sim \mathcal{D}, \, y \sim \pi_t(\cdot|x)} \left[ \frac{1}{|y|} \sum_{l=1}^{|y|} \frac{\pi(y_l \mid x, y_{<l})}{\pi_t(y_l \mid x, y_{<l})} A(x,y) \right], \tag{1}$$

where $y_l$ denotes the $l$-th token in the generated sequence $y$. We adopt this formulation to eliminate the off-policyness during updates, clipping heuristics, and additional hyperparameters, which would complicate our analysis in the following section. We note that this is a **clean** RL objective that has the same gradient as policy gradient and can be directly derived from the original RL objective of maximizing expected reward: $\mathbb{E}_{x \sim \mathcal{D}, \, y \sim \pi(\cdot|x)}[r(x,y)]$. The derivation is provided in Appendix A.

## 3 PRELIMINARY INVESTIGATIONS

In this section, we present a set of preliminary experiments that investigate the interplay between convergence, batch size, the number of prompts per batch, and the number of generations per prompt. We first define them in detail.

**Batch size**, denoted by $b$, refers to the total number of prompt–response pairs in a batch. In our purely on-policy setting, this number also corresponds to the total number of pairs used in a single update. The batch size is given by the product of the number of prompts and generations per prompt. Batch size directly affects the **generation time**, as larger batches require longer to generate.

**Number of prompts**, denoted by $m$, refers to the number of unique prompts in a batch. This quantity is closely related to the **prompt diversity**. Increasing the number of prompts improves the diversity of the batch, which in turn reduces gradient noise and stabilizes learning.

**Generations per prompt**, denoted by $n$, refers to the number of responses generated for each prompt. These responses are used to estimate the expected reward, which is used to compute the advantage. The number of generations per prompt is related to the **effective ratio**, defined as the proportion of samples in the batch with non-zero advantages, i.e., the proportion of samples that contribute meaningful gradient signals. Increasing $n$ improves the effective ratio. For example, for a particularly challenging prompt, if $n = 2$, both responses may be incorrect, leading to zero advantage and zero gradient under the objective in Eq. 1. In contrast, for $n = 16$ or 32, it is much more likely that at least one response is correct, resulting in a non-zero advantage and thus useful gradient updates. Therefore, increasing $n$ would result in a more accurate advantage estimation and a higher effective ratio.

**Convergence** is defined as the final training or validation reward achieved under a fixed compute and time budget (e.g., number of GPUs and wall-clock time). A method exhibits faster convergence if, under the same computational resources, it reaches a higher reward. Convergence is influenced by generation time, prompt diversity, and effective ratio. Reducing generation time enables more frequent updates, while increasing prompt diversity and effective ratio reduces noise in the gradient and leads to more stable and efficient training.

Overall, these quantities exhibit a natural trade-off. On the one hand, reducing generation time enables more frequent updates within a fixed time budget, allowing the model to train on new rollouts from improved policies. On the other hand, increasing the number of prompts and generations per prompt reduces gradient noise with a higher signal-to-noise ratio. In the following experiments, we perform comprehensive ablations with around 100K A100 GPU hours to identify the optimal balance between these competing factors.

## 3.1 OPTIMAL BATCH SIZE

**Experiment Setup.** We conduct experiments on both MATH (Hendrycks et al., 2021) and Deep-ScaleR (Luo et al., 2025) datasets. For MATH, we evaluate on the standard MATH500 split. For Deep-ScaleR, we include evaluations on MATH500, Minerva Math (Lewkowycz et al., 2022), Olympiad-Bench (He et al., 2024), as well as competition-level benchmarks including AMC 23, AIME 24, and AIME 25. We report results across four models, Qwen3-1.7B-base, Qwen3-4B-base, Qwen3-8B-base, and Llama3.2-3B-it, covering two model families and a range of sizes. All models are trained with a context length of $4,096$ tokens. We use a rule-based reward function based on `math-verify` (Hugging Face, 2024), which assigns a reward of +1 for correct ones and 0 for incorrect ones or generations that exceed the context limit. All experiments are implemented using the VERL (Sheng et al., 2025), a synchronous training setup that alternates between generation and optimization phases. For each batch size, we ablate to find the optimal learning rate with a total of 23 runs. Additional implementation and training details, including learning rate ablations, are provided in Appendix B.

The results for Qwen3-4B-Base are presented in Fig. 2 and 3, including training reward as a function of both training steps and wall-clock time (in hours), generation time per step using `vLLM` (Kwon et al., 2023), and test accuracy. For DeepScaleR runs, test accuracy is reported as the average across all six benchmarks. Full results are provided in Appendix C.

**Larger batch sizes converge faster in terms of steps.** As shown in Fig. 2 (Left), increasing the batch size consistently leads to faster convergence when measured in training steps. This is primarily because larger batches reduce gradient noise, allowing the use of higher learning rates without destabilizing training. The learning rates used in each configuration are listed in Tables 4 and 5.

**Generation time grows sublinearly at first, then linearly.** In Fig. 2 and 3, we plot generation time per step against batch size, alongside a dashed reference line representing linear growth (intersecting the origin). We observe that generation time initially increases sublinearly with batch size, and transitions to linear growth as batch size continues to increase. This behavior is expected: When the

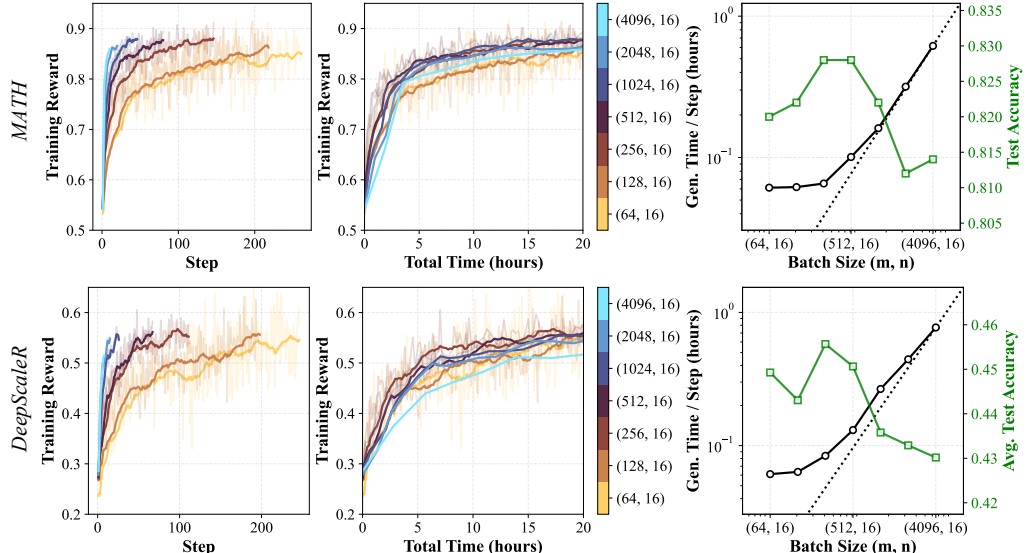

Figure 2: (Left / Middle) Training reward as a function of training steps and wall-clock time for Qwen3-4B-Base on MATH and DeepScaleR. The legend indicates the batch configuration in terms of (number of prompts $m$, generations per prompt $n$). (Right) Generation time per step and test accuracy across different batch sizes. The dashed line represents the linear increase that intercepts the origin and the generation time for the largest batch size. Both axes are in log scale. For key takeaways, refer to the paragraph headers in Section 3.1.

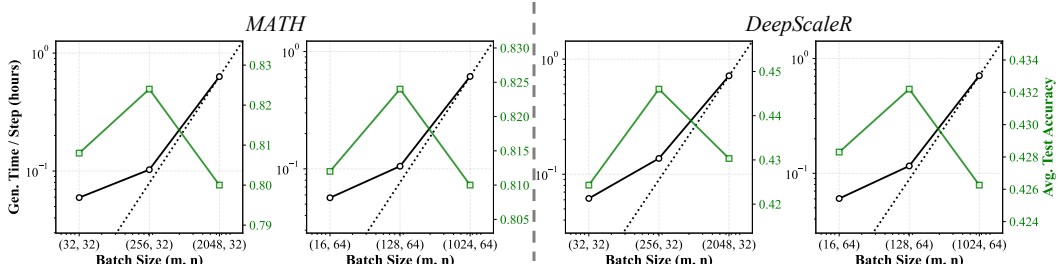

Figure 3: Generation time per step and test accuracy across different batch size combinations (number of prompts $m$, generations per prompt $n$) for Qwen3-4B-Base on MATH and DeepScaleR.

batch size is small, the generation time is dominated by the longest response in the batch. As batch size increases, compute utilization becomes the bottleneck, and generation time scales more linearly.

**Optimal batch size occurs at the transition point from sublinear to linear scaling.** From Fig. 2 (Middle / Right) and Fig. 3, there exists a sweet spot in batch size that yields the best convergence speed. Extremely small or large batch sizes lead to suboptimal performance. The optimal point for the fastest convergence tends to lie at the end of the sublinear regime and the beginning of the linear regime in generation time. Specifically, the optimal batch size in our setting is around 8K, achieved with combinations $(m, n) = (512, 16)$, $(256, 32)$, or $(128, 64)$. In other words, the optimal batch size remains fixed, regardless of how it is factorized into $m$ and $n$. We hypothesize that this sweet spot achieves a favorable balance: compared to smaller batch sizes, it can have linearly more generations with sublinear time growth; compared to larger batch sizes, it allows more frequent updates in the same amount of time. To ensure robustness, we validate this phenomenon across different model architectures and sizes, datasets, context lengths, hardware configurations, rollout engines (`vLLM` vs. `SGLang`), and batch configurations. Full results are provided in Appendix C. Having established an optimal batch size, the natural question is: *How should we determine the optimal decomposition into the number of prompts and generations per prompt?*

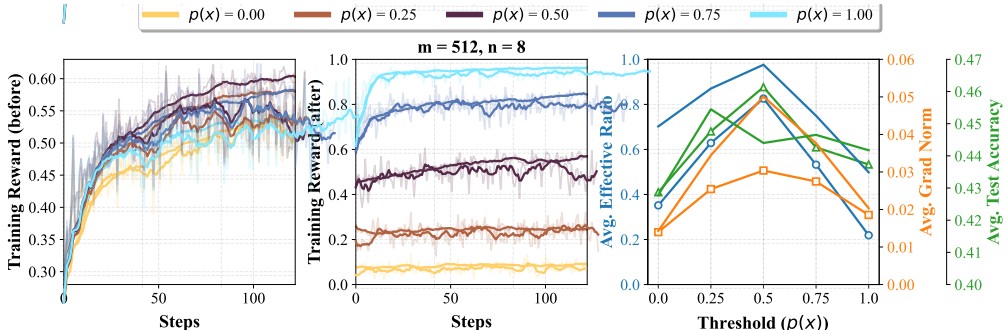

Figure 4: (Left) Training reward before downsampling in terms of step with number of prompts $m = 512$ and generations per prompt $n = 8$. (Middle) Training reward after downsampling. (Right) Average effective ratio and gradient norm over training steps, and average test accuracy of six benchmarks across different thresholds. For key takeaways, refer to Section 3.2.

## 3.2 OPTIMAL NUMBER OF PROMPTS AND GENERATIONS PER PROMPT

We hypothesize that the optimal decomposition of the batch size is closely tied to the difficulty of the prompts. Specifically, for extremely easy or difficult prompts, a larger number of generations ($n$) may be necessary to achieve a high effective ratio. In contrast, for prompts of intermediate difficulty ($p(x) \approx 0.5$), fewer generations may be sufficient.

**Experiment Setup.** We use DeepScaleR dataset and Qwen3-4B-Base, and train under different decompositions. To control prompt difficulty, for each batch we first sample $4m$ prompts and generate 4 responses for each prompt to estimate $p(x)$, similar to Zhang et al. (2025). We then perform greedy downsampling to select $m$ prompts that are closest to a specific difficulty threshold $p(x) \in \{0, 0.25, 0.5, 0.75, 1\}$, and sample $n$ generations per selected prompt for training. We are not reusing the 4 responses to train to avoid selection-induced bias, which keeps the ablation on $n$ comparable. We keep the total batch size fixed at $m \times n = 4096$ and ablate $n$ from 2 to 128. All other experimental configurations remain the same. Full results are shown in Appendix C.

**Downsampling successfully retains target-difficulty prompts.** As shown in Fig. 4 (Left / Middle), our downsampling procedure effectively retains prompts around the specified threshold. This validates the experimental design and ensures that training focuses on prompts of controlled difficulty.

**Higher $n$ improves effective ratio and $p(x) = 0.5$ has the highest effective ratio.** As shown in Fig. 4 (Right) and 5, increasing $n$ consistently improves the effective ratio, and prompts with $p(x) = 0.5$ achieve high effective ratios even with relatively small $n$. For example, the effective ratio for $n = 16$ at $p(x) = 0.5$ is already higher than any other thresholds even with $n = 128$.

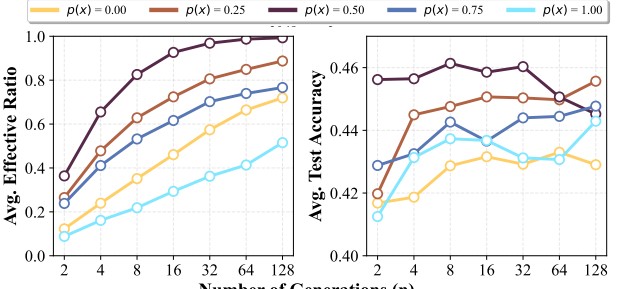

Figure 5: Average effective ratio over training steps and average test accuracy of six benchmarks under different thresholds $p(x)$ and generations per prompt $n$.

$p(x) = 0.5$ **has the highest gradient norm and test accuracy.** As shown in Fig. 4 (Right) and 5, training on prompts with $p(x) = 0.5$ yields the highest gradient norms and test accuracy. Interestingly, while increasing $n$ benefits test accuracy for other difficulty levels, we find that for $p(x) = 0.5$, accuracy actually degrades beyond $n = 32$. We suspect this is due to reduced prompt diversity (i.e., smaller $m$), which increases gradient noise despite higher per-prompt sampling. Conversely, based on the previous section, since there exists an optimal batch size, focusing on $p(x) = 0.5$ allows us to use a smaller $n$ and a higher $m$ which improves prompt diversity and also maintains a high effective ratio. In other words, we could have the best of both worlds (effective ratio and prompt diversity) with $p(x) = 0.5$. Full results, including ablations across all configurations, are provided in Appendix C, and a theoretical connection of the gradient norm and $p(x)$ is provided in Appendix D.

---

**Algorithm 1** PCL

---

**Require:** Number of prompts $m$, generations per prompt $n$, threshold $\tau$, sampling parameter $k$
1: Initialize policy $\pi_0$, value network $V^{\pi_{-1}}$
2: **for** $t = 0$ to $T - 1$ **do**
3:     Sample a batch with $km$ prompts: $\mathcal{D}_{km} = \{x^i\}_{i=1}^{km} \subset \mathcal{D}$.
4:     Select a batch of $m$ prompts using value model: $\mathcal{D}_m = \underset{S \subseteq \mathcal{D}_{km}, |S|=m}{\arg\min} \sum_{x \in S} \left| V^{\pi_{t-1}}(x) - \tau \right|$.
5:     Generate for the batch: $\mathcal{D}_m = \left\{ (x^i, \{y^{i,j}\}_{j=1}^n) \right\}_{i=1}^m$ where $y^{i,j} \overset{\text{iid}}{\sim} \pi_t(\cdot \mid x^i)$
6:     Update $\pi_t$ to $\pi_{t+1}$ using $\mathcal{D}_m$.
7:     Update $V^{\pi_{t-1}}$ to $V^{\pi_t}$ with loss in Eq. 2.
8: **end for**

---

## 4 PCL: PROMPT CURRICULUM LEARNING

The previous section demonstrates that prompts of intermediate difficulty ($p(x) \approx 0.5$) are the most sample-efficient for RL training. However, estimating the difficulty of each prompt using actual generations from the policy can be computationally expensive, as the generations for the filtered-out prompts are wasted. To address this issue, we propose a lightweight and efficient alternative: Prompt Curriculum Learning (**PCL**), which leverages a learned value model during online RL to estimate prompt difficulty using a single forward pass, significantly reducing computational overhead.

At training iteration $t$, we begin by sampling a pool of $km$ candidate prompts from the dataset where $k$ is a hyperparameter. For each prompt $x$, we use a value model to predict its expected reward $V(x)$, which approximates $p_\pi(x) = \mathbb{E}_{y \sim \pi(\cdot|x)}[r(x, y)]$. We then greedily select a subset of $m$ prompts whose predicted values are closest to a target difficulty threshold $\tau$ (defaulting to $0.5$), ensuring that the batch is focused on prompts of intermediate difficulty. For each selected prompt, we generate $n$ responses using the current policy and perform standard policy gradient updates. To update the value model, we only use the generated responses and minimize the prediction error between the estimated value $V(x)$ and the empirical average reward across the $n$ generations:

$$\sum_{i=1}^m \left( V(x^i) - \frac{1}{n} \sum_{j=1}^n r(x^i, y^{i,j}) \right)^2. \tag{2}$$

This allows us to improve the value model online, without requiring any additional rollouts. Since the value model only takes in the prompt as input which is typically less than 1K tokens in length for math, we find that both training and inference of the value model incur negligible cost (see Appendix I for a detailed breakdown). The full algorithm is summarized in Algorithm 1. Note that the value model $V$ in our algorithm is one step behind the policy $\pi$, which is acceptable since each update is small with $\pi_{t+1} \approx \pi_t$. We further discuss the alternatives in Section 7.

## 5 EXPERIMENTS

**Models & Datasets.** We use the same sets of models and datasets for experiments as Section 3. We use the same-sized model as the policy for the value model when running PCL. All runs use a 2-day time budget, except for Qwen3-8B-Base on DeepScaleR, which is trained for 3 days. We focus on $m = 512$ and $n = 16$ as it is one of the best combinations we found in terms of convergence. Unless otherwise noted, PCL uses $\tau = 0.5$ and $k = 4$. Similar to Wang et al. (2025c) and Zheng et al. (2025), we evaluate the model after training on every 4K prompts (8 steps), and report the performance of the checkpoint that obtains the best average performance.

**Baselines.** We compare PCL against five baselines. We include original **GRPO**, which performs no prompt filtering and uniformly samples prompts from the dataset. This serves as a standard baseline to assess the impact of filtering strategies. **Pre-filter** is a heuristic approach that leverages a fixed reference policy $\pi_{\text{ref}}$ to estimate prompt difficulty and filters out easy or hard prompts. **Dynamic-sampling (DS)** (Yu et al., 2025) uses $n$ rollouts per prompt to estimate $p_\pi$ for $km$ prompts and filters out prompts with $\hat{p}_\pi = 0$ or $1$. **SPEED** (Zhang et al., 2025) improves upon DS by first using $n_{\text{init}}$ rollouts to estimate where $n \geq n_{\text{init}}$. It then performs filtering and generates the remaining $n - n_{\text{init}}$

Table 1: **Results on MATH and DeepScaleR.** For each metric, the best-performing method is highlighted in **bold**, and the second-best is underlined. Time is the sum of training and generation time of the checkpoint that achieves the best average performance (excluding validation/checkpointing) in hours. For full DeepScaleR results across model sizes and families, refer to Appendix F.

| **MATH** | Qwen3-8B-Base | | Qwen3-4B-Base | | Qwen3-1.7B-Base | | Llama3.2-3B-it | |
| | MATH500 | Time | MATH500 | Time | MATH500 | Time | MATH500 | Time |
|---|---|---|---|---|---|---|---|---|
| $\pi_{\mathrm{ref}}$ | 72.4 | / | 65.6 | / | 55.4 | / | 42.6 | / |
| GRPO | 86.4 | 28.3 | 83.0 | 29.2 | 73.6 | 22.0 | 56.2 | **5.80** |
| Pre-filter | 84.8 | 17.1 | 81.6 | 27.1 | 73.4 | 13.5 | 55.4 | 7.47 |
| DS | 87.8 | 37.8 | 82.6 | 37.1 | **73.8** | 27.6 | 56.8 | 19.3 |
| SPEED | 81.2 | **4.25** | 78.8 | **6.75** | 70.2 | **1.93** | 42.6 | / |
| GRESO | 87.2 | 29.1 | 83.0 | 33.1 | 73.4 | 17.6 | 56.6 | 7.37 |
| PCL | **88.2** | 37.2 | **83.4** | 14.0 | **73.8** | 24.8 | **57.8** | 14.3 |

| | **DeepScaleR** | MATH500 | Olymp. | Minerva Avg@4 | AMC23 Avg@32 | AIME24 Avg@32 | AIME25 Avg@32 | Avg. | Time |
|---|---|---|---|---|---|---|---|---|---|
| | $\pi_{\mathrm{ref}}$ | 70.2 | 34.3 | 29.8 | 49.1 | 15.8 | 8.8 | 34.7 | / |
| | GRPO | 87.2 | 57.9 | 45.3 | 70.1 | 25.3 | 22.7 | 51.4 | 43.0 |
| Qwen3-8B-Base | Pre-filter | 86.4 | 54.6 | 44.2 | 69.8 | 26.9 | 22.6 | 50.7 | 67.4 |
| | DS | 87.2 | 55.3 | 45.7 | 71.5 | 24.9 | 24.2 | 51.5 | 69.5 |
| | SPEED | 82.4 | 46.4 | 40.3 | 66.6 | 21.1 | 15.7 | 45.5 | **19.3** |
| | PCL | 88.4 | 56.2 | 46.8 | 71.2 | 25.2 | 23.9 | **52.0** | 41.8 |

rollouts. **GRESO** (Zheng et al., 2025) keeps a dictionary of historical rewards based on generations from previous epochs and skips uninformative prompts using the dictionary. We tested GRESO on MATH but not on DeepScaleR, as DeepScaleR is large and limits the training to around 1 epoch under the compute budget which prevents the use of dictionary-based methods. DS, SPEED, and GRESO all keep sampling and generating until there is a full batch. Additional experiment details, including pseudo-codes and hyperparameters, are in Appendix E.

## 5.1 CONVERGENCE COMPARISON

**PCL either achieves the highest performance or requires significantly less training time to reach comparable performance.** The main results are summarized in Tables 1 (for full DeepScaleR results, refer to Appendix F). Compared to prior baselines, PCL consistently achieves the highest performance across all four models on the MATH dataset, and faster convergence at a similar or better accuracy on DeepScaleR. When training Qwen3-8B-Base on DeepScaleR, PCL converges **39.8%** faster than DS, the second-best method in terms of average accuracy. DS requires significantly more time to converge, as it performs generation for all $km$ prompts at each step with $n$ generations per prompt. SPEED's efficient implementation pre-generates $n_{\mathrm{init}}$ rollouts at an earlier step with an old policy and uses them at the current step, treating them as if sampled from the current policy. While this approach reduces generation cost for estimating $\hat{p}_\pi$, it introduces severe off-policyness. We observe that most of the SPEED runs crashed within a few hours, leading to lower convergence time as it would crash afterward. On the other hand, GRESO also suffers from a high degree of off-policyness where the historical estimates are based on outdated policies from the last epoch and may not reflect the current model's performance, especially when the dataset is large.

## 5.2 ANALYSIS & ABLATION

**PCL consistently achieves either a higher effective ratio or a lower generation time, while maintaining a focus on $p(x) = 0.5$ prompts.** To better understand the training dynamics of each method, we visualize the effective ratio, generation time per step, and training reward after filtering in Fig. 6 when training Qwen3-8B-Base on DeepScaleR. PCL consistently maintains a higher effective ratio compared to GRPO and Pre-filter. While DS and SPEED achieve an effective ratio of 1 due to resampling, they require significantly higher generation time, with relative increase of 105% and 81.8% for DS and SPEED respectively. The slightly higher generation time of PCL compared to GRPO and Pre-filter is that harder prompts require longer generations, and, when the average accuracy of the model on the training set is higher than 0.5, PCL focuses on harder prompts than those two methods. Interestingly, the effective ratio for Pre-filter starts higher than GRPO but quickly

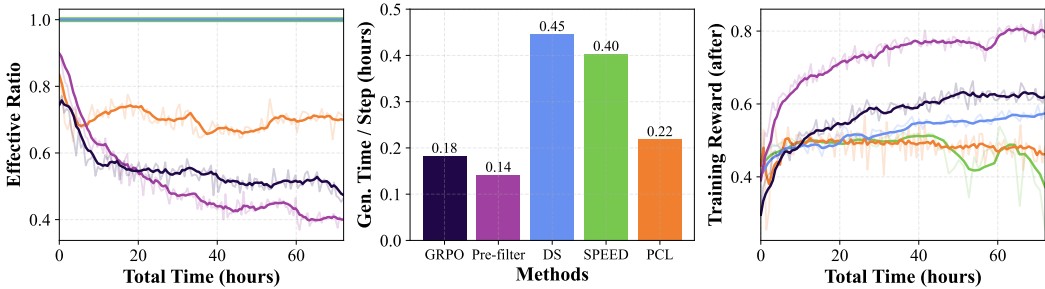

Figure 6: Experiment on DeepScaleR with Qwen3-8B-Base. (Left) Effective ratio w.r.t. training time across five methods. Refer to the middle plot for legend. (Middle) Average generation time per step throughout the training. (Right) Training reward after downsampling. PCL either has a higher effective ratio or a lower generation time, and is consistently training on $p(x) = 0.5$ prompts.

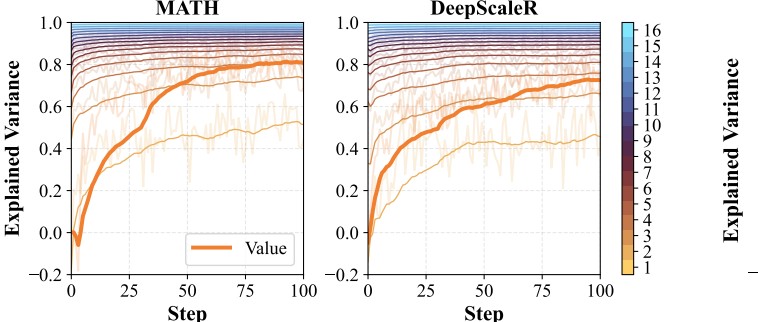
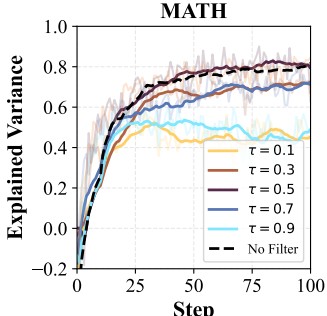

Figure 7: Explained variance of PCL's value model using 16 generations as the ground-truth difficulty ($p(x)$), and the explained variance using 1 to 16 generations to predict the difficulty ($\hat{p}(x)$) on MATH and DeepScaleR with two Qwen3-1.7B-Base models as policy and value model. The accuracy of the value model is similar to using around 3 generations to estimate.

Figure 8: Explained variance on MATH with Qwen3-1.7B-Base for PCL's value model with filtering on different thresholds ($\tau$) and without filtering.

drops below. This behavior comes from how Pre-filter selects prompts: it excludes very difficult ones based on $\pi_{\text{ref}}$. As the policy improves during training, many previously difficult prompts transition into the intermediate-difficulty range (e.g., $p(x) \approx 0.5$) for the current model. However, because these prompts were previously filtered out, they are never revisited, causing Pre-filter to keep training on easy prompts from the perspective of the current policy. In addition, as shown in Fig. 6 (Right), PCL consistently focuses on intermediate-difficulty prompts throughout training (the training reward of PCL after filtering stays closely to 0.5), whereas other methods gradually shift toward easier prompts as the policy improves which is suboptimal based on the findings in Section 3.

**The accuracy of the value model is similar to using 3 generations to estimate.** To investigate the prediction accuracy of the value model, we compute the explained variance using the average reward of 16 generations as the ground-truth difficulty $p(x)$. The explained variance is calculated as:

$$1 - \frac{\text{Var}\left(\{p(x^i) - V(x^i)\}_{i=1}^m\right)}{\text{Var}\left(\{p(x^i)\}_{i=1}^m\right)} \tag{3}$$

where Var denotes the variance. In addition, we also use the average reward of 1 to 16 generations as the predicted difficulty $\hat{p}(x)$ and compute their explained variance. The explained variances are computed on prompts randomly sampled from the dataset before filtering. The results on MATH and DeepScaleR with two Qwen3-1.7B-Base models as policy and value models are shown in Fig. 7. Since the prediction head of the value model is randomly initialized, the initial explained variance is very low. As training progresses, the value model improves steadily and achieves an explained variance comparable to using three rollouts per prompt for value estimation. Specifically, with $km = 2048$, generating $n = 3$ rollouts per prompt takes 288 seconds on MATH and 396 seconds on DeepScaleR per step. In contrast, training and inference with the value model require only 23.9 and 23.5 seconds respectively, achieving a 12.1× speedup on MATH and a 16.9× speedup on DeepScaleR. Results are visualized in Figure 1.

**The accuracy of the value model with filtering at $\tau = 0.5$ matches that of training without filtering.** One might expect the value model to suffer from filtering, as the training data is biased toward prompts with estimated difficulty near the threshold $\tau$, potentially limiting generalization. To investigate how the choice of threshold $\tau$ affects the accuracy of the value model, we ablate over $\tau \in \{0.1, 0.3, 0.5, 0.7, 0.9\}$. In addition, we train a baseline value model without any prompt filtering (i.e., GRPO but with a value model trained alongside the policy) using Qwen3-1.7B-Base for both the policy and value model on the MATH dataset. Results are presented in Fig. 8. We observe that the value model achieves the highest prediction accuracy when $\tau = 0.5$, with performance degrading as the threshold deviates further from 0.5 in either direction. Notably, the accuracy of the value model at $\tau = 0.5$ is comparable to the no-filtering baseline, despite training on a filtered subset of prompts. We hypothesize that filtering at $\tau = 0.5$ still captures a diverse set of reward outcomes, as it is the midpoint of the binary rewards. Moreover, if the average reward of the policy over the training data is not 0.5 (i.e., there is label imbalance), filtering around $\tau = 0.5$ may implicitly rebalance the data, thus improving generalization. In contrast, filtering with extreme $\tau$ values (e.g., $\tau = 0.1$ or $\tau = 0.9$) selects only very easy or very hard prompts, leading to severe label imbalance and reduced predictive accuracy. A deeper theoretical understanding of why $\tau = 0.5$ leads to such effective value model training is an interesting direction for future work.

**PCL progressively focuses on harder prompts during training, despite a fixed threshold of $\tau = 0.5$.** To better understand the training dynamics of PCL, we analyze how the difficulty of selected prompts evolves over time. Specifically, we use the initial reference policy $\pi_{\text{ref}}$ to generate 16 responses for each prompt in DeepScaleR and compute the average reward, which serves as a proxy for prompt difficulty (i.e., lower average rewards indicate harder prompts). During training on Qwen3-8B-Base with PCL, we log the average $\pi_{\text{ref}}$-based reward for the filtered prompts at each training step. The results are shown in Fig. 9. For methods that do not perform prompt filtering (GRPO and Pre-filter), this average remains nearly constant, as these methods uniformly sample from the dataset. In contrast, for methods that apply filtering (DS, SPEED, and PCL), we observe a consistent downward trend in the $\pi_{\text{ref}}$-based reward of selected prompts. This indicates that these methods focus on increasingly harder prompts as training progresses. Although PCL maintains a fixed difficulty threshold of $\tau = 0.5$, as the policy improves, previously hard prompts would now appear

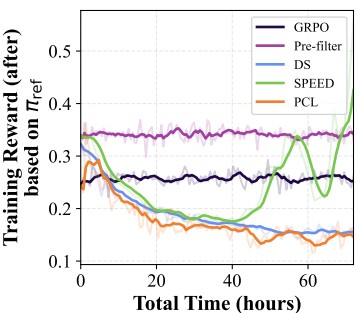

Figure 9: Training reward of PCL after filtering based on $\pi_{\text{ref}}$ w.r.t. training time with DeepScaleR and Qwen3-8B-Base. PCL progressively focuses on harder prompts during training, despite a fixed threshold of $\tau = 0.5$.

intermediate (i.e., $\tau \approx 0.5$), allowing PCL to continually shift toward more challenging examples.

## 6 RELATED WORK

**LLM Post-training.** Reinforcement learning (RL) has become a standard for post-training LLMs, including Reinforcement Learning from Human Feedback (RLHF) (Ouyang et al., 2022; Christiano et al., 2023; OpenAI, 2024a; Team, 2025a), enabling the LLMs to generate faithful and harmless responses that closely follow the instruction, and Reinforcement Learning with Verifiable Rewards (RLVR) (OpenAI, 2024b; Yang et al., 2024; DeepSeek-AI, 2025; Qwen, 2025; Lambert et al., 2025; Team, 2025b), improving model reasoning capabilities using verifiable rewards. These methods typically use algorithms include PPO (Schulman et al., 2017), GRPO (Shao et al., 2024), DR-GRPO (Liu et al., 2025), OREO (Wang et al., 2024), DQO (Ji et al., 2024), and VinePPO (Kazemnejad et al., 2025).

**Efficient RL for LLM Post-training.** Given the huge parameter size for LLMs, there is a large body of work recently focusing on developing more efficient algorithms and data selection methods to enable more efficient RL training for LLMs. Algorithmically, DPO (Rafailov et al., 2024), RAFT (Dong et al., 2023), REBEL (Gao et al., 2024), REFUEL (Gao et al., 2025), $A^\star$-PO (Brantley et al., 2025), RAFT++ (Xiong et al., 2025), RLOO (Ahmadian et al., 2024), and REINFORCE++ (Hu et al., 2025) are all trying to construct new objective functions that either reduces the number of models used (e.g. value model, reference model, reward model) or reduces the number of generations required for online RL. Another line of works (Xia et al., 2024; Muennighoff et al., 2025; Ye et al.,

2025a; Muldrew et al., 2024; Das et al., 2025; Wang et al., 2025c; Sun et al., 2025; Wang et al., 2025a; Lin et al., 2025) focuses on improving data selections by reducing the amount of training data to be more sample efficient. DAPO (Yu et al., 2025) and VAPO (Yue et al., 2025) resample and keep generating until the effective ratio of the batch is 1 during each step of RL training. However, the generations for a prompt that are either all correct or incorrect are wasted. SPEED (Zhang et al., 2025) improves on top of these methods by using a smaller number of generations to estimate the effective ratio and only generate the rest of the generations if the existing ones are not all correct or incorrect. GRESO (Zheng et al., 2025) and MoPPS (Qu et al., 2025), on the other hand, avoids rollouts by using a dictionary or a Beta posterior and samples based on the historical rewards from the previous epoch. However, they would suffer from off-policyness especially when the dataset is large.

Our method is the combination of the best of both worlds where PCL directly avoids costly rollouts and also is on-policy. Our method is closely related to a classic class of machine learning techniques, Curriculum Learning (Bengio et al., 2009). Previous works have explored curriculum learning for LLM post-training (Lee et al., 2024; Wen et al., 2025; Shi et al., 2025) by either training on progressively harder prompts ordered before training or focusing on certain difficulty range on the fly during RL. Our work falls in this group by always focusing on intermediate difficulty prompts for the current policy.

**Curriculum and Active Learning.** More broadly, our approach connects to curriculum and active learning methods that adapt the training distribution over examples. In supervised learning, Mindermann et al. (2022) prioritize examples based on loss on a held-out set, while Ash et al. (2020) selects diverse and uncertain examples using gradient embeddings. Kawaguchi & Lu (2020) analyzes how sample ordering affects convergence under Ordered SGD. These methods assume access to per-example gradients or labels on a static dataset, whereas we operate in an online RLVR setting where each example is a long response that cannot be determined before generating. In RL, Parker-Holder et al. (2022) and Ye et al. (2025b) study environment curricula by evolving tasks through regret-based environment design or asymmetric self-play. Under their minimax-regret formulations, the optimal teacher concentrates on the hardest solvable tasks, in contrast to our empirical finding that convergence is fastest when training on prompts of intermediate difficulty. PCL adapts and extends ideas from curriculum and active learning to the specific constraints of large-scale, on-policy RLVR-style LLM post-training.

# 7 DISCUSSIONS & CONCLUSION

PCL accelerates RL post-training by targeting two findings from our study: (1) there exists an optimal total batch size at the transition between sublinear and linear generation-time scaling, and (2) prompts of intermediate difficulty ($p(x) \approx 0.5$) yield the highest gradient signal and sample efficiency. It trains a value model online to identify such prompts, avoiding the wasted rollouts of generation-based filtering (DS, SPEED) and the off-policyness of dictionary-based methods (GRESO). PCL either achieves the highest performance or requires significantly less training time to reach comparable performance. We include a discussion on limitations in Appendix J.

While our experiments focus on binary correctness rewards, PCL naturally extends to non-binary scalar rewards. Since the value model $V(x)$ estimates $\mathbb{E}_{y \sim \pi(\cdot|x)}[r(x,y)]$, non-binary $r(x,y)$ only changes its range and the meaning of the target threshold $\tau$. In addition, we note that PCL alternates updates between the policy and the value model, meaning that $V^{\pi_t}$ is always one step behind the current policy $\pi_{t+1}$. In practice, this lag does not hinder performance, as the per-step policy updates are small with $\pi_t \approx \pi_{t+1}$. Theoretically, Wang et al. (2025b) provably shows that the ranking of prompt difficulties is stable under small perturbations between $\pi_t$ and $\pi_{t+1}$. We also experimented with using importance sampling to correct for this lag by reweighting based on $\pi_{t+1}(y \mid x)/\pi_t(y \mid x)$, but it does not improve the accuracy of the value model and computing $\pi_{t+1}(y \mid x)$ is computationally expensive as $y$ is thousands of tokens long.

We highlight that prompt filtering methods rely on an implicit assumption of prompt-level generalization: training on a selected subset of prompts will improve performance on the filtered-out ones. For example, PCL assumes that training on intermediate-difficulty prompts leads to improvements on both easier and harder prompts, while DS and SPEED assume that gradually solving not-too-hard prompts enables the model to eventually handle harder ones. While this assumption holds in domains like math where problems often share structural similarities, it may not generalize to other domains.

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

# Appendix

## CONTENTS

# A    PROBLEM SETUP DETAILS

Let $x$ denote a prompt (e.g., a math question), and let $y$ denote a sampled solution of length $|y|$ generated autoregressively from a policy $\pi$, i.e., $y \sim \pi(\cdot \mid x)$. We assume a binary reward function $r(x,y) \in \{0,1\}$, where $r(x,y) = 1$ if the final answer in $y$ is correct and $0$ otherwise. Our goal is to learn a parameterized policy $\pi_\theta$ that maximizes the expected reward over a dataset $\mathcal{D}$ of prompts:

$$J(\theta) = \mathbb{E}_{x\sim\mathcal{D},\, y\sim\pi_\theta(\cdot|x)}[r(x,y)]. \tag{4}$$

Following the standard REINFORCE derivation (Williams, 1992), the gradient of this objective can be written as $\nabla_\theta J(\theta) = \mathbb{E}_{x,y}\left[r(x,y)\nabla_\theta \log \pi_\theta(y \mid x)\right]$.

To reduce the variance of this estimator, it is common to subtract a baseline function that depends only on the prompt $x$, which does not change the optimum of the policy gradient (Kool et al., 2019; Richter et al., 2020; Zhu et al., 2023; Shao et al., 2024). In this work, we use the expected reward under the current policy, $\mathbb{E}_{y'\sim\pi_\theta(\cdot|x)}[r(x,y')]$, as the baseline, which is standard in LLM post-training (Shao et al., 2024; DeepSeek-AI, 2025; Yu et al., 2025; Liu et al., 2025). Since the reward is binary, we define $p_{\pi_\theta}(x) := \mathbb{E}_{y\sim\pi_\theta(\cdot|x)}[r(x,y)]$ as the probability of generating a correct answer, and $A(x,y) := r(x,y) - p_{\pi_\theta}(x)$ as the advantage. The policy gradient can be expressed as $\nabla_\theta J(\theta) = \mathbb{E}_{x\sim\mathcal{D},\, y\sim\pi_\theta(\cdot|x)}\left[A(x,y)\nabla_\theta \log \pi_\theta(y \mid x)\right]$.

In practice, LLMs are trained with multiple updates on generations produced by some old policy $\pi_{\theta_{\text{old}}}$ and the training is often stabilized using techniques such as PPO-style clipping (Schulman et al., 2017; Shao et al., 2024; Xiong et al., 2025). However, we focus on a **purely on-policy** setting, where each gradient step is followed by the collection of fresh rollouts. Specifically, at each iteration $t$, we perform a single gradient step to maximize:

$$J(\theta) = \mathbb{E}_{x\sim\mathcal{D},\, y\sim\pi_\theta(\cdot|x)}[A(x,y)\log\pi_\theta(y \mid x)]. \tag{5}$$

Note that the above objective has the same gradient as:

$$J(\theta) = \mathbb{E}_{x\sim\mathcal{D},\, y\sim\pi_{\theta_t}(\cdot|x)}\left[A(x,y)\frac{\pi_\theta(y \mid x)}{\pi_{\theta_t}(y \mid x)}\right], \tag{6}$$

since we are purely on-policy and $\pi_{\theta_t}$ is the policy before the update and also serves as the sampling distribution.

Given the autoregressive nature of LLMs, we further decompose the objective into a token-level form, treating each token as an individual action:

$$J(\theta) = \mathbb{E}_{x\sim\mathcal{D},\, y\sim\pi_\theta(\cdot|x)}\left[A(x,y)\log\left(\prod_{l=1}^{|y|}\pi_\theta(y_l \mid x, y_{<l})\right)\right] \tag{7}$$

$$= \mathbb{E}_{x\sim\mathcal{D},\, y\sim\pi_\theta(\cdot|x)}\left[A(x,y)\sum_{l=1}^{|y|}\log\pi_\theta(y_l \mid x, y_{<l})\right], \tag{8}$$

where $y_l$ denotes the $l$-th token in the generated sequence. Similarly, the above objective has the same gradient as:

$$J(\theta) = \mathbb{E}_{x\sim\mathcal{D},\, y\sim\pi_{\theta_t}(\cdot|x)}\left[A(x,y)\sum_{l=1}^{|y|}\frac{\pi_\theta(y_l \mid x, y_{<l})}{\pi_{\theta_t}(y_l \mid x, y_{<l})}\right]. \tag{9}$$

Normalize by the length of $y$, we arrive at

$$J(\theta) = \mathbb{E}_{x\sim\mathcal{D},\, y\sim\pi_{\theta_t}(\cdot|x)}\left[\frac{1}{|y|}A(x,y)\sum_{l=1}^{|y|}\frac{\pi_\theta(y_l \mid x, y_{<l})}{\pi_{\theta_t}(y_l \mid x, y_{<l})}\right]. \tag{10}$$

This objective corresponds to a purely on-policy variant of GRPO (Shao et al., 2024; DeepSeek-AI, 2025), without KL regularization to a fixed reference policy $\pi_{\text{ref}}$ (Yu et al., 2025) and without standard deviation-based advantage regularization (Liu et al., 2025). We adopt this formulation to eliminate the off-policyness during updates, clipping heuristics, and additional hyperparameters. This results in a **clean** experimental setup that is directly derived from the original RL objective in Eq. 4.

# B PRELIMINARY INVESTIGATION DETAILS

## B.1 DATASET DETAILS

Table 2: Dataset split, maximum prompt length, and maximum generation length

| Dataset | Huggingface Dataset Card | Train - Val | Prompt Length | Generation Length |
|---|---|---|---|---|
| MATH | DigitalLearningGmbH/MATH-lighteval | 7.5k - 5k | 1,024 | 4,096 |
| DeepScaleR | agentica-org/DeepScaleR-Preview-Dataset | 40.3k - / | 1,024 | 4,096 |

Table 3: Model prompt format

| Model Family | Prompt Format |
|---|---|
| Qwen (Base) | **{prompt}** Let's think step by step and output the final answer within \boxed{}. |
| Llama (Instruct) | <\|begin_of_text\|><\|start_header_id\|>system<\|end_header_id\|>Cutting Knowledge Date: December 2023 Today Date: 26 Jul 2024<\|eot_id\|><\|start_header_id\|>user<\|end_header_id\|>**{prompt}** Let's think step by step and output the final answer within \boxed{}. <\|eot_id\|><\|start_header_id\|>assistant<\|end_header_id\|> |

## B.2 MODEL DETAILS

We perform **full parameter** training on 8 A100 GPUs using Qwen3-1.7B-Base (model card: Qwen/Qwen3-1.7B-Base), Qwen3-4B-Base (model card: Qwen/Qwen3-4B-Base), Qwen3-8B-Base (model card: Qwen/Qwen3-8B-Base), and Llama3.2-3B-it (model card: meta-llama/Llama-3.2-3B-Instruct).

## B.3 REWARD DETAILS

We use a rule-based reward function based on the correctness of the response with math-verify, assigning +1 for correct answers and 0 for incorrect ones or generations that exceed the context length. Recent studies (Chen et al., 2025) have proposed incorporating format-based rules into reward calculations to encourage models to follow specific output formats. However, in our experiments, we observed no significant difference in performance with or without such format-based rewards. Therefore, for simplicity, we exclude them from our implementation.

## B.4 EVALUATION DETAILS

Following prior work (Zeng et al., 2025), we evaluate model performance on a suite of standard mathematical reasoning benchmarks, including MATH500 (Hendrycks et al., 2021), Minerva Math (Lewkowycz et al., 2022), and OlympiadBench (He et al., 2024), as well as competition-level benchmarks such as AMC 2023, AIME 2024, and AIME 2025.

For smaller-scale datasets, we report results using the average reward across multiple generations. Specifically, for Minerva Math, we report `Avg@4`; for AMC 2023, AIME 2024, and AIME 2025, we report `Avg@32`.

For MATH experiments, we use decoding parameters `top_k` = 20, `temperature` = 0.6, and `top_p` = 0.95. For DeepScaleR experiments, we use `top_k` = −1 (i.e., disabled), `temperature` = 0.6, and `top_p` = 0.95.

## B.5 COMPLETE LIST OF EXPERIMENTS

The learning rate for each batch size is tuned on a logarithmic scale using the Qwen3-8B-Base model. For all other models, we adopt the corresponding optimal learning rate found for Qwen3-8B-Base. The complete list of all the experiments is provided below with the chosen learning rate highlighted in **bold**.

Table 4: Complete List of Experiments for Math

| Model | #Prompts ($m$) | #Generations ($n$) | Context Length | Num Workers | Engine | Batch Size ($b$) | LR |
|---|---|---|---|---|---|---|---|
| Qwen3-8B-base | 64 | 16 | 4096 | 8 | VLLM | 1024 | 1E-6/**2E-6** |
| | 128 | 16 | 4096 | 8 | VLLM | 2048 | 1E-6/**2E-6**/5E-6/1E-5 |
| | 256 | 16 | 4096 | 8 | VLLM | 4096 | 2E-6/**4E-6**/8E-6 |
| | 512 | 16 | 4096 | 8 | VLLM | 8192 | 4E-6/**8E-6**/1.6E-5 |
| | 1024 | 16 | 4096 | 8 | VLLM | 16384 | 4E-6/**8E-6**/1.6E-5 |
| | 2048 | 16 | 4096 | 8 | VLLM | 32768 | 4E-6/8E-6/**1.6E-5**/3.2E-5 |
| | 4096 | 16 | 4096 | 8 | VLLM | 65536 | 8E-6/1.6E-5/**3.2E-5**/6.4E-5 |
| Qwen3-4B-base | 64 | 16 | 4096 | 8 | VLLM | 1024 | 2.00E-06 |
| | 128 | 16 | 4096 | 8 | VLLM | 2048 | 2.00E-06 |
| | 256 | 16 | 4096 | 8 | VLLM | 4096 | 4.00E-06 |
| | 512 | 16 | 4096 | 8 | VLLM | 8192 | 8.00E-06 |
| | 1024 | 16 | 4096 | 8 | VLLM | 16384 | 8.00E-06 |
| | 2048 | 16 | 4096 | 8 | VLLM | 32768 | 1.60E-05 |
| | 4096 | 16 | 4096 | 8 | VLLM | 65536 | 3.20E-05 |
| Qwen3-1.7B-base | 64 | 16 | 4096 | 8 | VLLM | 1024 | 2.00E-06 |
| | 128 | 16 | 4096 | 8 | VLLM | 2048 | 2.00E-06 |
| | 256 | 16 | 4096 | 8 | VLLM | 4096 | 4.00E-06 |
| | 512 | 16 | 4096 | 8 | VLLM | 8192 | 8.00E-06 |
| | 1024 | 16 | 4096 | 8 | VLLM | 16384 | 8.00E-06 |
| | 2048 | 16 | 4096 | 8 | VLLM | 32768 | 1.60E-05 |
| | 4096 | 16 | 4096 | 8 | VLLM | 65536 | 3.20E-05 |
| Llama3.2-3B-it | 64 | 16 | 4096 | 8 | VLLM | 1024 | 2.00E-06 |
| | 128 | 16 | 4096 | 8 | VLLM | 2048 | 2.00E-06 |
| | 256 | 16 | 4096 | 8 | VLLM | 4096 | 4.00E-06 |
| | 512 | 16 | 4096 | 8 | VLLM | 8192 | 8.00E-06 |
| | 1024 | 16 | 4096 | 8 | VLLM | 16384 | 8.00E-06 |
| | 2048 | 16 | 4096 | 8 | VLLM | 32768 | 8.00E-06 |
| | 4096 | 16 | 4096 | 8 | VLLM | 65536 | 1.20E-05 |
| Qwen3-4B-base | 32 | 32 | 4096 | 8 | VLLM | 1024 | 2.00E-06 |
| | 256 | 32 | 4096 | 8 | VLLM | 8192 | 8.00E-06 |
| | 2048 | 32 | 4096 | 8 | VLLM | 65536 | 3.20E-05 |
| | 16 | 64 | 4096 | 8 | VLLM | 1024 | 2.00E-06 |
| | 128 | 64 | 4096 | 8 | VLLM | 8192 | 8.00E-06 |
| | 1024 | 64 | 4096 | 8 | VLLM | 65536 | 3.20E-05 |

Table 5: Complete List of Experiments for DeepScaleR

| Model | #Prompts ($m$) | #Generations ($n$) | Context Length | Num Workers | Engine | Batch Size | LR |
|---|---|---|---|---|---|---|---|
| Qwen3-8B-base | 64 | 16 | 4096 | 8 | VLLM | 1024 | 1E-6/**2E-6** |
| | 128 | 16 | 4096 | 8 | VLLM | 2048 | 1E-6/**2E-6**/5E-6/1E-5/2E-5 |
| | 256 | 16 | 4096 | 8 | VLLM | 4096 | 2E-6/**4E-6**/8E-6 |
| | 512 | 16 | 4096 | 8 | VLLM | 8192 | 2E-6/**4E-6**/6E-6 |
| | 1024 | 16 | 4096 | 8 | VLLM | 16384 | 4E-6/**8E-6**/1.2E-5/1.6E-5 |
| | 2048 | 16 | 4096 | 8 | VLLM | 32768 | 8E-6/**1.2E-5**/1.6E-5 |
| | 4096 | 16 | 4096 | 8 | VLLM | 65536 | 8E-6/**1.2E-5**/1.6E-5 |
| Qwen3-4B-base | 64 | 16 | 4096 | 8 | VLLM | 1024 | 2.00E-06 |
| | 128 | 16 | 4096 | 8 | VLLM | 2048 | 2.00E-06 |
| | 256 | 16 | 4096 | 8 | VLLM | 4096 | 4.00E-06 |
| | 512 | 16 | 4096 | 8 | VLLM | 8192 | 4.00E-06 |
| | 1024 | 16 | 4096 | 8 | VLLM | 16384 | 8.00E-06 |
| | 2048 | 16 | 4096 | 8 | VLLM | 32768 | 1.20E-05 |
| | 4096 | 16 | 4096 | 8 | VLLM | 65536 | 1.20E-05 |
| Qwen3-1.7B-base | 64 | 16 | 4096 | 8 | VLLM | 1024 | 2.00E-06 |
| | 128 | 16 | 4096 | 8 | VLLM | 2048 | 2.00E-06 |
| | 256 | 16 | 4096 | 8 | VLLM | 4096 | 4.00E-06 |
| | 512 | 16 | 4096 | 8 | VLLM | 8192 | 4.00E-06 |
| | 1024 | 16 | 4096 | 8 | VLLM | 16384 | 8.00E-06 |
| | 2048 | 16 | 4096 | 8 | VLLM | 32768 | 1.20E-05 |
| | 4096 | 16 | 4096 | 8 | VLLM | 65536 | 1.20E-05 |
| Llama3.2-3B-it | 64 | 16 | 4096 | 8 | VLLM | 1024 | 2.00E-06 |
| | 128 | 16 | 4096 | 8 | VLLM | 2048 | 2.00E-06 |
| | 256 | 16 | 4096 | 8 | VLLM | 4096 | 4.00E-06 |
| | 512 | 16 | 4096 | 8 | VLLM | 8192 | 4.00E-06 |
| | 1024 | 16 | 4096 | 8 | VLLM | 16384 | 8.00E-06 |
| | 2048 | 16 | 4096 | 8 | VLLM | 32768 | 8.00E-06 |
| | 4096 | 16 | 4096 | 8 | VLLM | 65536 | 1.20E-05 |
| Qwen3-4B-base | 32 | 32 | 4096 | 8 | VLLM | 1024 | 2.00E-06 |
| | 256 | 32 | 4096 | 8 | VLLM | 8192 | 4.00E-06 |
| | 2048 | 32 | 4096 | 8 | VLLM | 65536 | 1.20E-05 |
| | 16 | 64 | 4096 | 8 | VLLM | 1024 | 2.00E-06 |
| | 128 | 64 | 4096 | 8 | VLLM | 8192 | 4.00E-06 |
| | 1024 | 64 | 4096 | 8 | VLLM | 65536 | 1.20E-05 |
| Qwen3-4B-base | 64 | 16 | 8192 | 8 | VLLM | 1024 | 2.00E-06 |
| | 128 | 16 | 8192 | 8 | VLLM | 2048 | 2.00E-06 |
| | 256 | 16 | 8192 | 8 | VLLM | 4096 | 4.00E-06 |
| | 512 | 16 | 8192 | 8 | VLLM | 8192 | 4.00E-06 |
| | 1024 | 16 | 8192 | 8 | VLLM | 16384 | 8.00E-06 |
| | 2048 | 16 | 8192 | 8 | VLLM | 32768 | 1.20E-05 |
| | 4096 | 16 | 8192 | 8 | VLLM | 65536 | 1.20E-05 |
| Qwen3-4B-base | 16 | 16 | 4096 | 1 | VLLM | 256 | 1.00E-06 |
| | 32 | 16 | 4096 | 1 | VLLM | 512 | 1.00E-06 |
| | 64 | 16 | 4096 | 1 | VLLM | 1024 | 2.00E-06 |
| | 128 | 16 | 4096 | 1 | VLLM | 2048 | 2.00E-06 |
| | 256 | 16 | 4096 | 1 | VLLM | 4096 | 4.00E-06 |
| | 512 | 16 | 4096 | 1 | VLLM | 8192 | 4.00E-06 |
| | 1024 | 16 | 4096 | 1 | VLLM | 16384 | 8.00E-06 |
| Qwen3-4B-base | 16 | 16 | 4096 | 8 | SGLang | 256 | 1.00E-06 |
| | 32 | 16 | 4096 | 8 | SGLang | 512 | 1.00E-06 |
| | 64 | 16 | 4096 | 8 | SGLang | 1024 | 2.00E-06 |
| | 128 | 16 | 4096 | 8 | SGLang | 2048 | 2.00E-06 |
| | 256 | 16 | 4096 | 8 | SGLang | 4096 | 4.00E-06 |
| | 512 | 16 | 4096 | 8 | SGLang | 8192 | 4.00E-06 |
| | 1024 | 16 | 4096 | 8 | SGLang | 16384 | 8.00E-06 |
| | 2048 | 16 | 4096 | 8 | SGLang | 32768 | 1.20E-05 |
| | 4096 | 16 | 4096 | 8 | SGLang | 65536 | 1.20E-05 |

Table 6: Complete List of Experiments for DeepScaleR (cont.)

| Model | #Prompts ($m$) | #Generations ($n$) | Context Length | Num Workers | Engine | Batch Size | LR |
|---|---|---|---|---|---|---|---|
| | 32 | 128 | 4096 | 8 | VLLM | 4096 | 4.00E-06 |
| | 64 | 64 | 4096 | 8 | VLLM | 4096 | 4.00E-06 |
| | 128 | 32 | 4096 | 8 | VLLM | 4096 | 4.00E-06 |
| | 256 | 16 | 4096 | 8 | VLLM | 4096 | 4.00E-06 |
| Qwen3-4B-Base & $p(x) = 0$ | 512 | 8 | 4096 | 8 | VLLM | 4096 | 4.00E-06 |
| | 1024 | 4 | 4096 | 8 | VLLM | 4096 | 4.00E-06 |
| | 2048 | 2 | 4096 | 8 | VLLM | 4096 | 4.00E-06 |
| | 32 | 128 | 4096 | 8 | VLLM | 4096 | 4.00E-06 |
| | 64 | 64 | 4096 | 8 | VLLM | 4096 | 4.00E-06 |
| | 128 | 32 | 4096 | 8 | VLLM | 4096 | 4.00E-06 |
| | 256 | 16 | 4096 | 8 | VLLM | 4096 | 4.00E-06 |
| Qwen3-4B-Base & $p(x) = 0.25$ | 512 | 8 | 4096 | 8 | VLLM | 4096 | 4.00E-06 |
| | 1024 | 4 | 4096 | 8 | VLLM | 4096 | 4.00E-06 |
| | 2048 | 2 | 4096 | 8 | VLLM | 4096 | 4.00E-06 |
| | 32 | 128 | 4096 | 8 | VLLM | 4096 | 4.00E-06 |
| | 64 | 64 | 4096 | 8 | VLLM | 4096 | 4.00E-06 |
| | 128 | 32 | 4096 | 8 | VLLM | 4096 | 4.00E-06 |
| | 256 | 16 | 4096 | 8 | VLLM | 4096 | 4.00E-06 |
| Qwen3-4B-Base & $p(x) = 0.5$ | 512 | 8 | 4096 | 8 | VLLM | 4096 | 4.00E-06 |
| | 1024 | 4 | 4096 | 8 | VLLM | 4096 | 4.00E-06 |
| | 2048 | 2 | 4096 | 8 | VLLM | 4096 | 4.00E-06 |
| | 32 | 128 | 4096 | 8 | VLLM | 4096 | 4.00E-06 |
| | 64 | 64 | 4096 | 8 | VLLM | 4096 | 4.00E-06 |
| | 128 | 32 | 4096 | 8 | VLLM | 4096 | 4.00E-06 |
| | 256 | 16 | 4096 | 8 | VLLM | 4096 | 4.00E-06 |
| Qwen3-4B-Base & $p(x) = 0.75$ | 512 | 8 | 4096 | 8 | VLLM | 4096 | 4.00E-06 |
| | 1024 | 4 | 4096 | 8 | VLLM | 4096 | 4.00E-06 |
| | 2048 | 2 | 4096 | 8 | VLLM | 4096 | 4.00E-06 |
| | 32 | 128 | 4096 | 8 | VLLM | 4096 | 4.00E-06 |
| | 64 | 64 | 4096 | 8 | VLLM | 4096 | 4.00E-06 |
| | 128 | 32 | 4096 | 8 | VLLM | 4096 | 4.00E-06 |
| | 256 | 16 | 4096 | 8 | VLLM | 4096 | 4.00E-06 |
| Qwen3-4B-Base & $p(x) = 1$ | 512 | 8 | 4096 | 8 | VLLM | 4096 | 4.00E-06 |
| | 1024 | 4 | 4096 | 8 | VLLM | 4096 | 4.00E-06 |
| | 2048 | 2 | 4096 | 8 | VLLM | 4096 | 4.00E-06 |

# C    PRELIMINARY INVESTIGATION COMPLETE RESULTS

## C.1    COMPLETE RESULTS FOR SECTION 3.1

### C.1.1    RESULTS WITH VARYING $m$

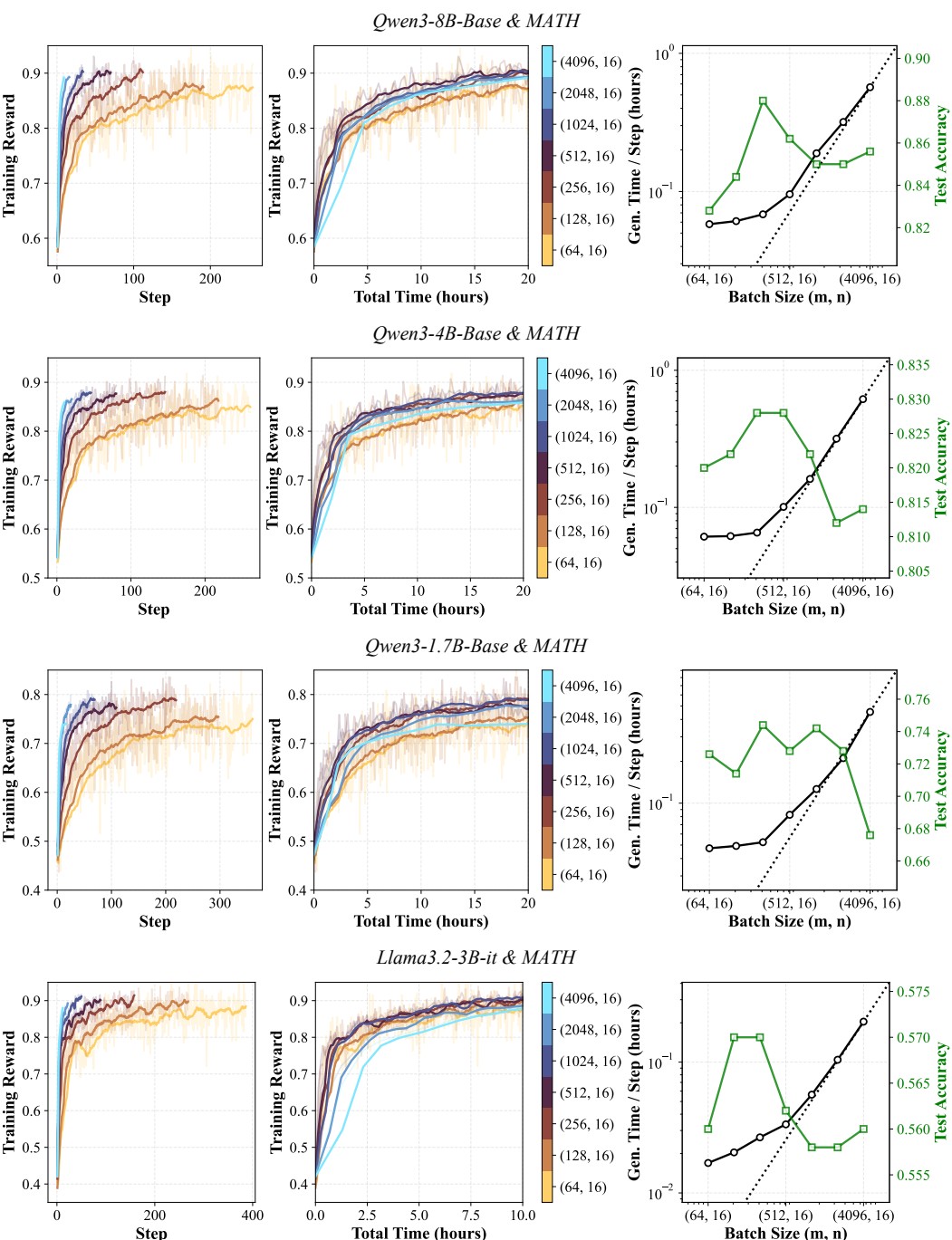

Figure 10: Results for all four models on MATH with $n = 16$. (Left / Middle) Training reward as a function of training steps and wall-clock time. The legend indicates the batch configuration in terms of (number of prompts $m$, generations per prompt $n$). (Right) Generation time per step and test accuracy across different batch sizes. The dashed line represents the linear increase that intercepts the origin and the generation time for the largest batch size. Both axes are in log scale.

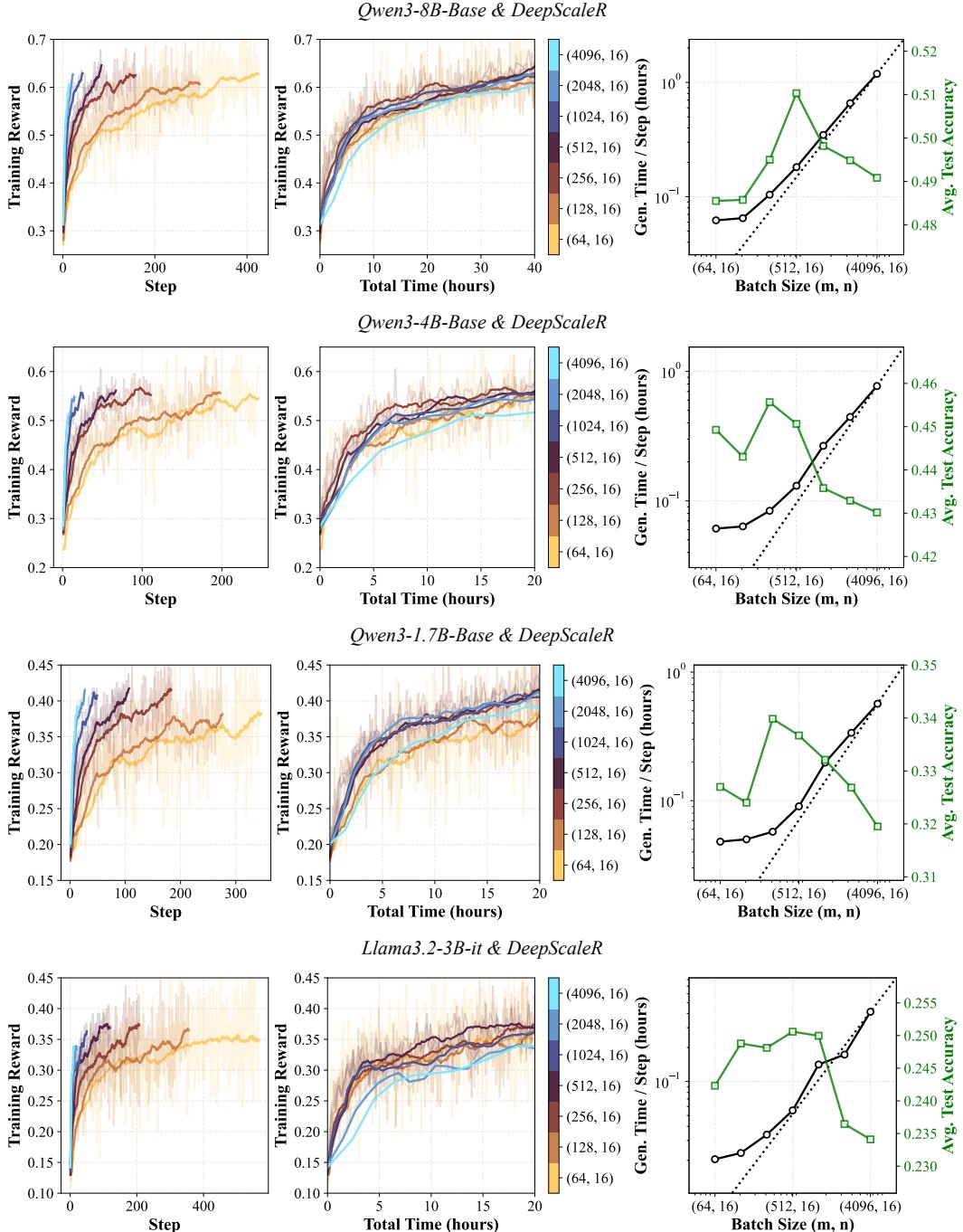

Figure 11: Results for all four models on DeepScaleR with $n = 16$. (Left / Middle) Training reward as a function of training steps and wall-clock time. The legend indicates the batch configuration in terms of (number of prompts $m$, generations per prompt $n$). (Right) Generation time per step and test accuracy across different batch sizes. The dashed line represents the linear increase that intercepts the origin and the generation time for the largest batch size. Both axes are in log scale.

Table 7: Detailed Results for Fig. 11.

| Model | $m$ | MATH500 | Olymp. | Minerva Avg@4 | AMC23 Avg@32 | AIME24 Avg@32 | AIME25 Avg@32 | Avg. |
|---|---|---|---|---|---|---|---|---|
| Qwen3-8B-Base | $\pi_{\text{ref}}$ | 70.2 | 34.3 | 29.8 | 49.1 | 15.8 | 8.8 | 34.7 |
| | 4096 | 85.8 | 52.2 | 43.0 | 70.9 | 21.5 | 21.1 | 49.1 |
| | 2048 | 85.2 | 53.4 | 43.9 | 66.6 | 26.4 | 21.5 | 49.5 |
| | 1024 | 85.6 | 54.9 | 45.9 | 70.5 | 22.8 | 19.3 | 49.8 |
| | 512 | 87.2 | 55.8 | 44.1 | 71.8 | 26.1 | 21.1 | 51.0 |
| | 256 | 85.0 | 57.4 | 40.4 | 66.3 | 24.9 | 22.9 | 49.5 |
| | 128 | 85.6 | 53.9 | 42.0 | 67.8 | 22.3 | 19.9 | 48.6 |
| | 64 | 85.2 | 54.7 | 42.1 | 70.6 | 21.4 | 17.3 | 48.6 |
| Qwen3-4B-Base | $\pi_{\text{ref}}$ | 65.8 | 34.4 | 26.9 | 47.3 | 10.9 | 7.1 | 32.1 |
| | 4096 | 80.6 | 45.8 | 39.7 | 59.8 | 16.4 | 15.8 | 43.0 |
| | 2048 | 83.2 | 48.4 | 39.2 | 57.0 | 16.0 | 15.9 | 43.3 |
| | 1024 | 81.6 | 46.1 | 40.2 | 59.3 | 18.1 | 16.1 | 43.6 |
| | 512 | 84.0 | 49.7 | 38.8 | 62.9 | 17.1 | 17.9 | 45.1 |
| | 256 | 82.8 | 48.1 | 40.3 | 66.3 | 17.8 | 18.1 | 45.6 |
| | 128 | 83.8 | 46.0 | 42.6 | 59.5 | 18.2 | 15.6 | 44.3 |
| | 64 | 83.2 | 48.2 | 39.7 | 64.1 | 17.6 | 16.8 | 44.9 |
| Qwen3-1.7B-Base | $\pi_{\text{ref}}$ | 57.0 | 23.9 | 21.8 | 29.0 | 3.8 | 1.1 | 22.8 |
| | 4096 | 69.8 | 35.2 | 29.0 | 40.7 | 9.1 | 8.0 | 32.0 |
| | 2048 | 70.2 | 34.3 | 31.2 | 42.0 | 12.2 | 6.2 | 32.7 |
| | 1024 | 72.2 | 36.2 | 29.7 | 41.8 | 12.4 | 7.0 | 33.2 |
| | 512 | 71.8 | 37.1 | 30.1 | 44.2 | 12.7 | 6.1 | 33.7 |
| | 256 | 72.6 | 35.6 | 31.5 | 46.9 | 10.1 | 7.2 | 34.0 |
| | 128 | 68.4 | 35.2 | 30.0 | 43.3 | 10.9 | 6.7 | 32.4 |
| | 64 | 70.2 | 36.5 | 30.0 | 40.8 | 11.2 | 7.5 | 32.7 |
| Llama3.2-3B-it | $\pi_{\text{ref}}$ | 42.8 | 12.3 | 13.8 | 19.7 | 4.6 | 0.4 | 15.6 |
| | 4096 | 55.2 | 20.0 | 21.1 | 31.6 | 11.9 | 0.6 | 23.4 |
| | 2048 | 55.8 | 19.3 | 21.2 | 30.9 | 13.8 | 0.9 | 23.6 |
| | 1024 | 57.8 | 22.8 | 21.2 | 34.8 | 12.1 | 1.2 | 25.0 |
| | 512 | 58.0 | 22.3 | 22.7 | 30.0 | 15.8 | 1.6 | 25.1 |
| | 256 | 57.6 | 21.8 | 22.6 | 32.0 | 14.5 | 0.4 | 24.8 |
| | 128 | 55.6 | 22.7 | 22.2 | 34.8 | 13.9 | 0.1 | 24.9 |
| | 64 | 56.8 | 20.9 | 25.5 | 31.8 | 10.2 | 0.2 | 24.2 |

### C.1.2 RESULTS WITH VARYING $m$ AND $n$

Table 8: Detailed DeepScaleR Results for Fig. 12.

| Model | $m$ | $n$ | MATH500 | Olymp. | Minerva Avg@4 | AMC23 Avg@32 | AIME24 Avg@32 | AIME25 Avg@32 | Avg. |
|---|---|---|---|---|---|---|---|---|---|
| Qwen3-4B-Base | $\pi_{\text{ref}}$ | | 65.8 | 34.4 | 26.9 | 47.3 | 10.9 | 7.1 | 32.1 |
| | 32 | 32 | 80.4 | 47.5 | 37.4 | 57.4 | 17.5 | 14.4 | 42.4 |
| | 256 | 32 | 83.2 | 49.1 | 38.6 | 63.4 | 17.3 | 16.0 | 44.6 |
| | 2048 | 32 | 81.4 | 46.3 | 39.8 | 57.6 | 17.6 | 15.5 | 43.0 |
| | 16 | 64 | 81.6 | 47.8 | 39.2 | 58.4 | 15.7 | 14.3 | 42.8 |
| | 128 | 64 | 83.4 | 44.4 | 41.5 | 60.0 | 17.0 | 13.1 | 43.2 |
| | 1024 | 64 | 80.8 | 48.7 | 39.2 | 57.0 | 16.4 | 13.8 | 42.6 |

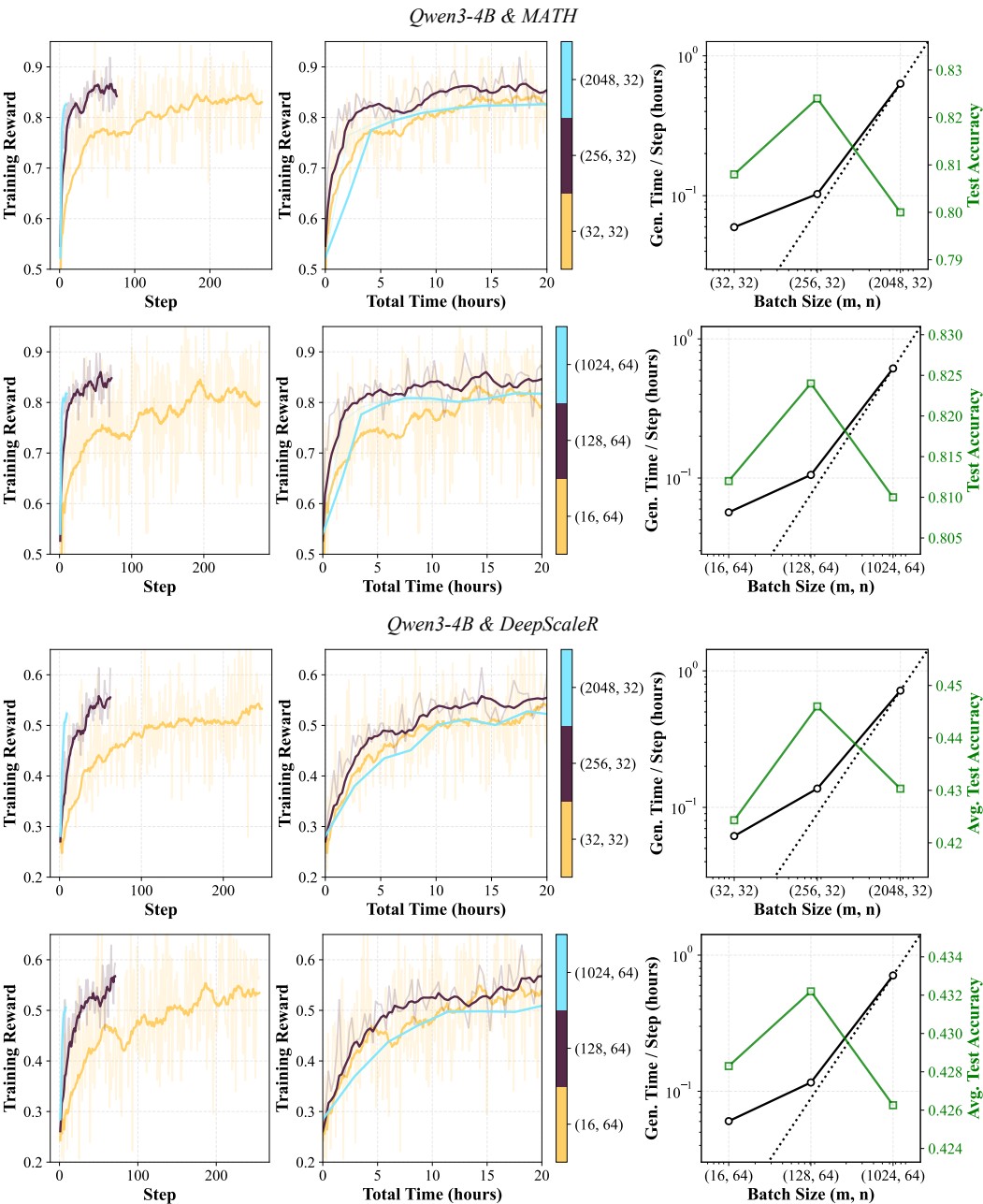

Figure 12: Results for Qwen3-4B on MATH and DeepScaleR with $n = 32$ and $64$. (Left / Middle) Training reward as a function of training steps and wall-clock time. The legend indicates the batch configuration in terms of (number of prompts $m$, generations per prompt $n$). (Right) Generation time per step and test accuracy across different batch sizes. The dashed line represents the linear increase that intercepts the origin and the generation time for the largest batch size. Both axes are in log scale.

## C.1.3 RESULTS WITH A DIFFERENT CONTEXT LENGTH

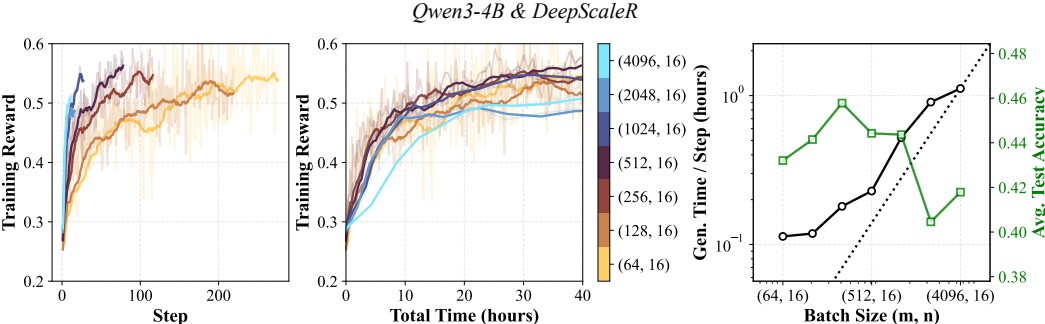

Figure 13: Results for Qwen3-4B on DeepScaleR with context length 8192 (other results are using 4096 context length). (Left / Middle) Training reward as a function of training steps and wall-clock time. The legend indicates the batch configuration in terms of (number of prompts $m$, generations per prompt $n$). (Right) Generation time per step and test accuracy across different batch sizes. The dashed line represents the linear increase that intercepts the origin and the generation time for the largest batch size. Both axes are in log scale.

Table 9: Detailed DeepScaleR Results for Fig. 13.

| Model | $m$ | Context Len. | MATH500 | Olymp. | Minerva Avg@4 | AMC23 Avg@32 | AIME24 Avg@32 | AIME25 Avg@32 | Avg. |
|---|---|---|---|---|---|---|---|---|---|
| | $\pi_{\text{ref}}$ | | 65.8 | 34.4 | 26.9 | 47.3 | 10.9 | 7.1 | 32.1 |
| | 64 | 8K | 80.4 | 47.9 | 39.2 | 60.7 | 15.9 | 15.1 | 43.2 |
| | 128 | 8K | 83.0 | 50.6 | 38.4 | 62.3 | 14.2 | 16.4 | 44.1 |
| | 256 | 8K | 82.8 | 50.4 | 42.2 | 62.0 | 19.3 | 18.0 | 45.8 |
| Qwen3-4B-Base | 512 | 8K | 81.2 | 49.6 | 41.0 | 65.3 | 17.0 | 12.5 | 44.4 |
| | 1024 | 8K | 80.8 | 46.6 | 39.7 | 63.7 | 16.6 | 18.9 | 44.4 |
| | 2048 | 8K | 77.6 | 44.4 | 37.5 | 54.5 | 14.2 | 14.6 | 40.5 |
| | 4096 | 8K | 79.8 | 45.3 | 38.9 | 58.4 | 14.7 | 13.8 | 41.8 |

### C.1.4 RESULTS WITH A DIFFERENT HARDWARE CONFIGURATION

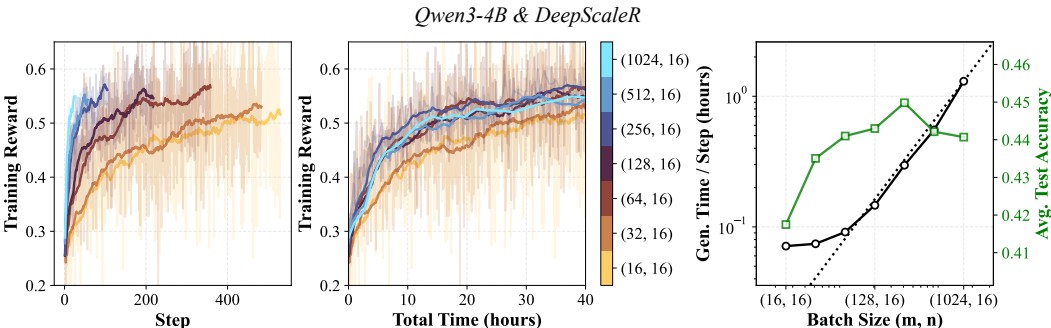

Figure 14: Results for Qwen3-4B on DeepScaleR with only 1 rollout worker with 8 GPUs (other results are using 8 rollout workers, 1 per GPU). (Left / Middle) Training reward as a function of training steps and wall-clock time. The legend indicates the batch configuration in terms of (number of prompts $m$, generations per prompt $n$). (Right) Generation time per step and test accuracy across different batch sizes. The dashed line represents the linear increase that intercepts the origin and the generation time for the largest batch size. Both axes are in log scale.

Table 10: Detailed DeepScaleR Results for Fig. 14.

| Model | $m$ | Num. Worker | MATH500 | Olymp. | Minerva Avg@4 | AMC23 Avg@32 | AIME24 Avg@32 | AIME25 Avg@32 | Avg. |
|---|---|---|---|---|---|---|---|---|---|
| | $\pi_{\mathrm{ref}}$ | | 65.8 | 34.4 | 26.9 | 47.3 | 10.9 | 7.1 | 32.1 |
| | 16 | 1 | 78.4 | 47.2 | 39.2 | 58.0 | 15.3 | 12.4 | 41.7 |
| | 32 | 1 | 81.8 | 47.0 | 39.0 | 61.3 | 17.1 | 14.9 | 43.5 |
| | 64 | 1 | 83.6 | 49.4 | 40.6 | 58.7 | 15.9 | 16.4 | 44.1 |
| Qwen3-4B-Base | 128 | 1 | 82.4 | 48.7 | 39.2 | 62.7 | 17.2 | 15.7 | 44.3 |
| | 256 | 1 | 82.8 | 49.6 | 38.6 | 62.6 | 18.9 | 17.5 | 45.0 |
| | 512 | 1 | 81.6 | 47.6 | 40.7 | 62.1 | 17.7 | 15.5 | 44.2 |
| | 1024 | 1 | 80.4 | 47.8 | 40.4 | 60.1 | 18.9 | 16.9 | 44.1 |

## C.1.5 RESULTS WITH A DIFFERENT INFERENCE ENGINE

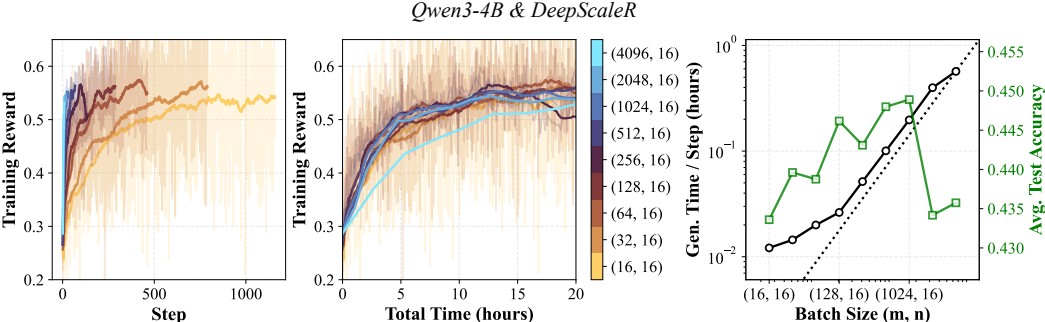

Figure 15: Results for Qwen3-4B on DeepScaleR with SGLang (other results are using VLLM). (Left / Middle) Training reward as a function of training steps and wall-clock time. The legend indicates the batch configuration in terms of (number of prompts $m$, generations per prompt $n$). (Right) Generation time per step and test accuracy across different batch sizes. The dashed line represents the linear increase that intercepts the origin and the generation time for the largest batch size. Both axes are in log scale.

Table 11: Detailed DeepScaleR Results for Fig. 15.

| Model | $m$ | Inference Eng. | MATH500 | Olymp. | Minerva Avg@4 | AMC23 Avg@32 | AIME24 Avg@32 | AIME25 Avg@32 | Avg. |
|---|---|---|---|---|---|---|---|---|---|
| Qwen3-4B-Base | $\pi_{\text{ref}}$ | | 65.8 | 34.4 | 26.9 | 47.3 | 10.9 | 7.1 | 32.1 |
| | 16 | SGLang | 81.8 | 48.5 | 37.3 | 58.0 | 17.1 | 17.4 | 43.4 |
| | 32 | SGLang | 81.2 | 49.9 | 40.0 | 60.9 | 17.0 | 14.9 | 44.0 |
| | 64 | SGLang | 81.6 | 50.9 | 39.8 | 60.2 | 15.3 | 15.4 | 43.9 |
| | 128 | SGLang | 84.0 | 48.8 | 39.7 | 62.9 | 15.2 | 17.1 | 44.6 |
| | 256 | SGLang | 81.8 | 49.4 | 40.0 | 60.4 | 16.6 | 17.7 | 44.3 |
| | 512 | SGLang | 82.2 | 50.3 | 40.9 | 63.2 | 17.0 | 15.2 | 44.8 |
| | 1024 | SGLang | 81.2 | 51.3 | 41.0 | 62.9 | 16.8 | 16.1 | 44.9 |
| | 2048 | SGLang | 81.2 | 46.0 | 40.6 | 61.0 | 17.3 | 14.4 | 43.4 |
| | 4096 | SGLang | 82.0 | 47.8 | 39.8 | 61.9 | 16.2 | 13.8 | 43.6 |

## C.2 COMPLETE RESULTS FOR SECTION 3.2

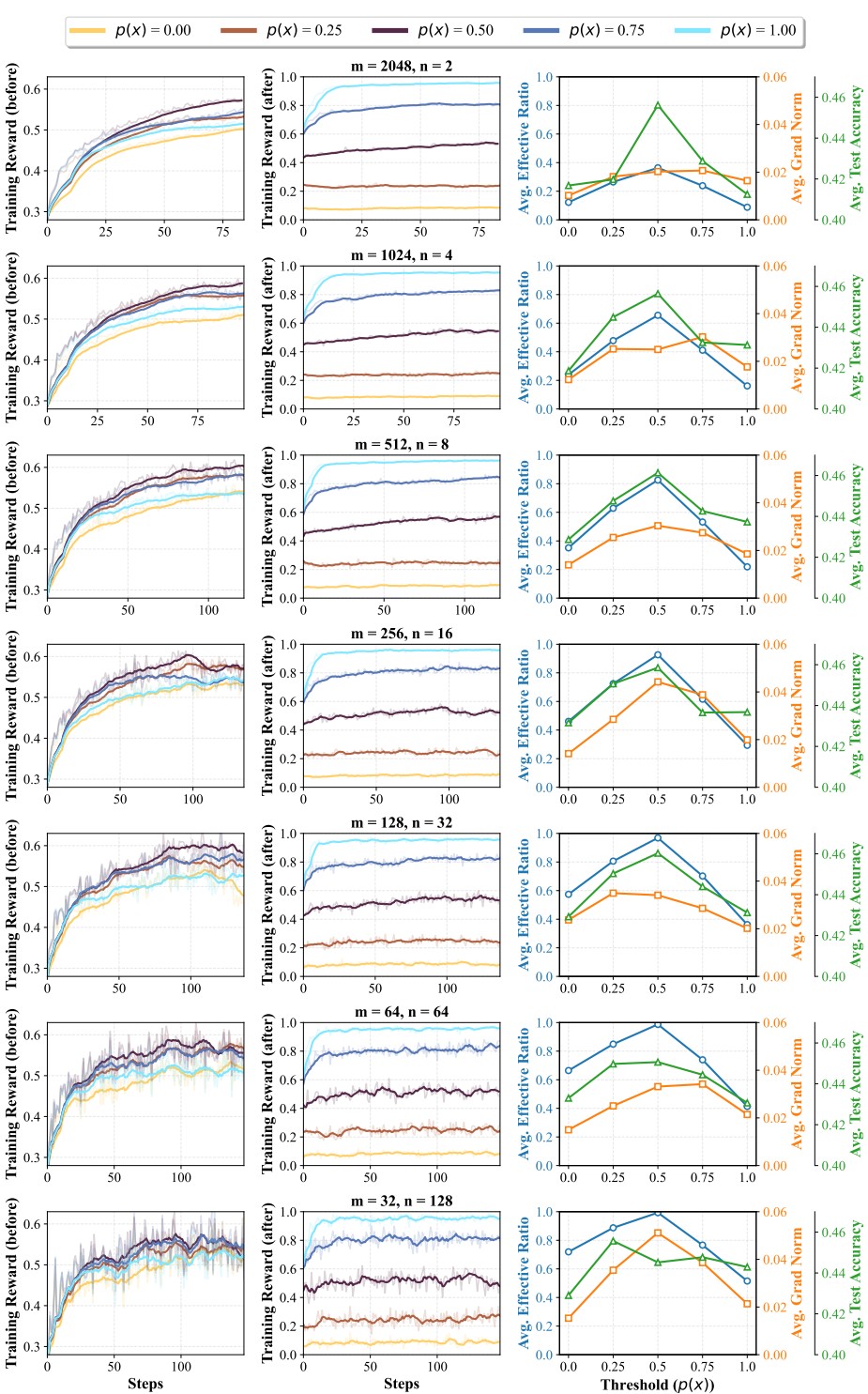

Figure 16: Results for Qwen3-4B on DeepScaleR with different $p(x)$ under different decompositions, grouped by number of prompts $m$ and generations per prompt $n$. (Left) Training reward before downsampling in terms of step. (Middle) Training reward after downsampling. (Right) Average effective ratio, gradient norm, and test accuracy across different thresholds.

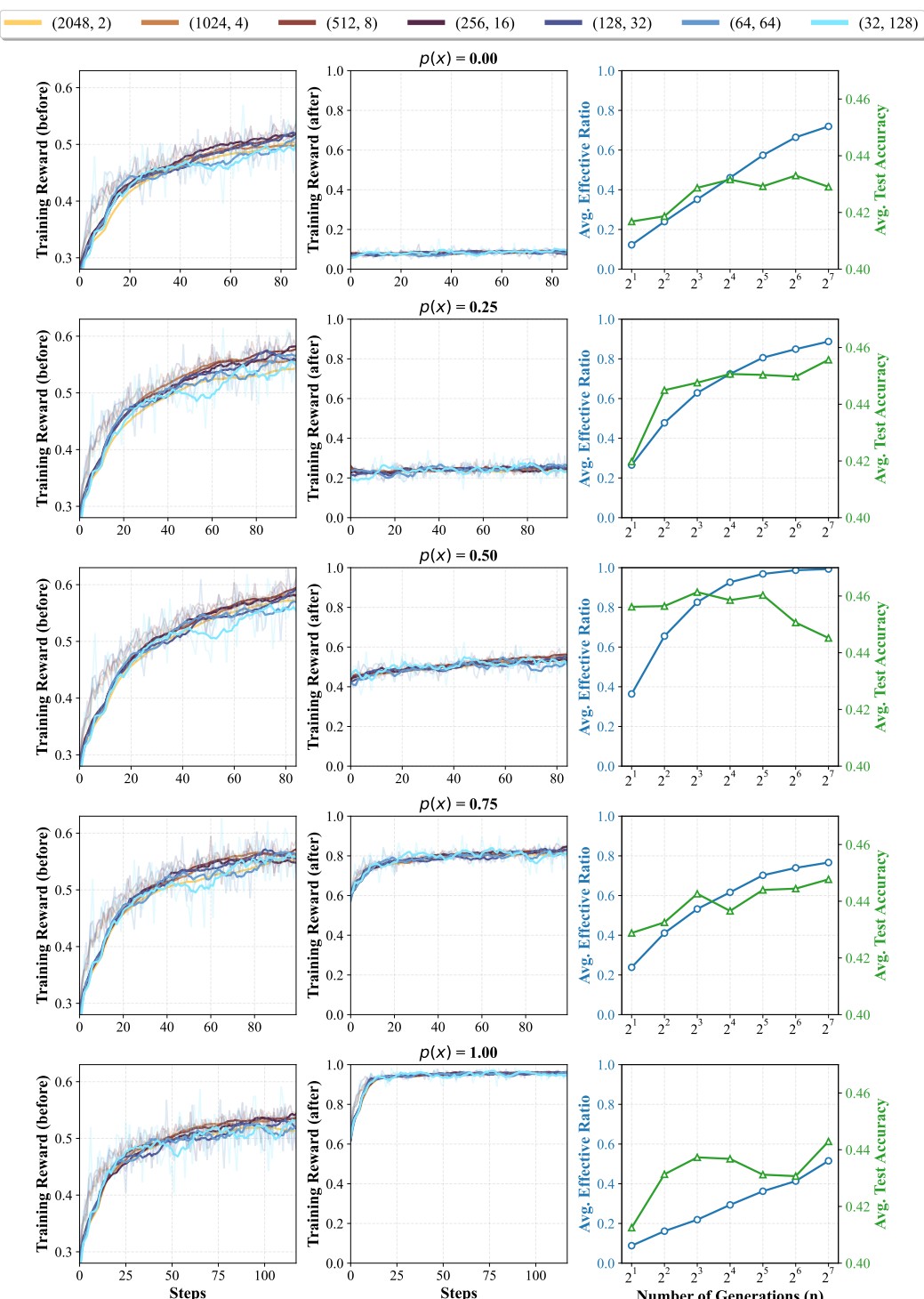

Figure 17: Results for Qwen3-4B on DeepScaleR with different $p(x)$ under different decompositions (number of prompts $m$, generations per prompt $n$), grouped by $p(x)$. (Left) Training reward before downsampling in terms of step. (Middle) Training reward after downsampling. (Right) Average effective ratio, gradient norm, and test accuracy across different thresholds.

Table 12: Detailed DeepScaleR Results for Fig. 16 and 17.

| $p(x)$ | $m$ | $n$ | MATH500 | Olymp. | Minerva Avg@4 | AMC23 Avg@32 | AIME24 Avg@32 | AIME25 Avg@32 | Avg. |
|---|---|---|---|---|---|---|---|---|---|
| / | $\pi_{\mathrm{ref}}$ | | 65.8 | 34.4 | 26.9 | 47.3 | 10.9 | 7.1 | 32.1 |
| 0 | 2048 | 2 | 82.8 | 44.8 | 39.4 | 54.3 | 15.3 | 13.4 | 41.7 |
| | 1024 | 4 | 81.8 | 44.7 | 38.4 | 54.8 | 17.1 | 14.5 | 41.9 |
| | 512 | 8 | 82.6 | 48.1 | 40.0 | 57.5 | 14.7 | 14.4 | 42.9 |
| | 256 | 16 | 83.4 | 46.6 | 39.4 | 58.0 | 16.7 | 14.9 | 43.2 |
| | 128 | 32 | 79.8 | 45.1 | 38.1 | 59.4 | 18.2 | 16.9 | 42.9 |
| | 64 | 64 | 83.4 | 47.3 | 40.7 | 59.3 | 15.4 | 13.6 | 43.3 |
| | 32 | 128 | 81.0 | 49.1 | 38.5 | 54.8 | 17.7 | 16.2 | 42.9 |
| 0.25 | 2048 | 2 | 80.6 | 46.9 | 39.4 | 58.3 | 15.5 | 11.1 | 42.0 |
| | 1024 | 4 | 84.2 | 48.2 | 39.9 | 61.6 | 16.6 | 16.5 | 44.5 |
| | 512 | 8 | 81.8 | 49.3 | 40.1 | 61.2 | 20.7 | 15.5 | 44.8 |
| | 256 | 16 | 82.8 | 47.0 | 38.5 | 63.2 | 19.6 | 19.3 | 45.1 |
| | 128 | 32 | 81.0 | 50.1 | 40.1 | 63.0 | 21.6 | 14.4 | 45.0 |
| | 64 | 64 | 83.4 | 49.1 | 41.6 | 58.8 | 19.1 | 17.8 | 45.0 |
| | 32 | 128 | 84.0 | 49.1 | 40.0 | 61.9 | 19.5 | 19.0 | 45.6 |
| 0.5 | 2048 | 2 | 83.2 | 50.7 | 40.0 | 60.9 | 19.2 | 19.7 | 45.6 |
| | 1024 | 4 | 81.4 | 53.7 | 40.5 | 63.7 | 17.2 | 17.3 | 45.6 |
| | 512 | 8 | 84.2 | 50.1 | 40.3 | 64.6 | 21.5 | 16.1 | 46.1 |
| | 256 | 16 | 82.8 | 49.7 | 41.5 | 62.4 | 19.5 | 19.3 | 45.9 |
| | 128 | 32 | 84.4 | 51.8 | 39.8 | 64.4 | 20.1 | 15.7 | 46.0 |
| | 64 | 64 | 83.6 | 48.8 | 39.4 | 61.9 | 19.9 | 16.8 | 45.1 |
| | 32 | 128 | 81.6 | 45.7 | 40.8 | 62.0 | 20.1 | 16.9 | 44.5 |
| 0.75 | 2048 | 2 | 81.0 | 48.7 | 39.0 | 56.6 | 17.5 | 14.6 | 42.9 |
| | 1024 | 4 | 82.0 | 49.7 | 38.6 | 58.3 | 16.5 | 14.5 | 43.3 |
| | 512 | 8 | 81.6 | 47.6 | 40.6 | 60.9 | 16.6 | 18.2 | 44.3 |
| | 256 | 16 | 81.4 | 46.1 | 39.1 | 64.0 | 17.2 | 14.2 | 43.7 |
| | 128 | 32 | 81.6 | 50.0 | 40.3 | 63.6 | 15.0 | 15.9 | 44.4 |
| | 64 | 64 | 81.0 | 50.0 | 41.1 | 61.9 | 15.1 | 17.6 | 44.4 |
| | 32 | 128 | 82.0 | 49.3 | 40.6 | 62.0 | 17.7 | 17.1 | 44.8 |
| 1 | 2048 | 2 | 78.6 | 46.6 | 41.6 | 55.8 | 14.3 | 10.6 | 41.3 |
| | 1024 | 4 | 80.8 | 48.8 | 38.3 | 60.2 | 16.7 | 14.0 | 43.1 |
| | 512 | 8 | 82.4 | 45.7 | 39.9 | 59.0 | 18.8 | 16.7 | 43.7 |
| | 256 | 16 | 82.0 | 46.7 | 40.6 | 58.8 | 18.4 | 15.5 | 43.7 |
| | 128 | 32 | 81.4 | 48.2 | 39.3 | 57.6 | 17.0 | 15.2 | 43.1 |
| | 64 | 64 | 81.2 | 46.9 | 40.7 | 56.8 | 18.8 | 14.1 | 43.1 |
| | 32 | 128 | 82.4 | 45.5 | 39.7 | 62.9 | 16.6 | 18.6 | 44.3 |

# D    CONNECTION BETWEEN $p(x)$ AND GRADIENT MAGNITUDE

We now analyze the squared norm of the gradient in Eq. (6):

$$\|\nabla_\theta J_x(\theta)\|_2 = \|\mathbb{E}_{y \sim \pi_\theta(\cdot|x)}[A(x,y)\nabla_\theta \log \pi_\theta(y|x)]\|_2. \tag{11}$$

Using Jensen inequality, we have:

$$\|\mathbb{E}_{y \sim \pi_\theta(\cdot|x)}[A(x,y)\nabla_\theta \log \pi_\theta(y|x)]\|_2 \leq \mathbb{E}_{y \sim \pi_\theta(\cdot|x)}[\|A(x,y)\nabla_\theta \log \pi_\theta(y|x)\|_2] \tag{12}$$

$$= \mathbb{E}_{y \sim \pi_\theta(\cdot|x)}[|A(x,y)| \, \|\nabla_\theta \log \pi_\theta(y|x)\|_2]. \tag{13}$$

Apply Cauchy-Schwarz:

$$\mathbb{E}_{y \sim \pi_\theta(\cdot|x)}[|A(x,y)| \, \|\nabla_\theta \log \pi_\theta(y|x)\|_2] \leq \sqrt{\mathbb{E}_{y \sim \pi_\theta(\cdot|x)}[A(x,y)^2]}\sqrt{\mathbb{E}_{y \sim \pi_\theta(\cdot|x)}[\|\nabla_\theta \log \pi_\theta(y|x)\|_2^2]}. \tag{14}$$

Now let's derive for $\mathbb{E}_{y \sim \pi_\theta(\cdot|x)}[A(x,y)^2]$:

$$\mathbb{E}_{y \sim \pi_\theta(\cdot|x)}[A(x,y)^2] = \mathbb{E}_{y \sim \pi_\theta(\cdot|x)}[(r(x,y) - p_{\pi_\theta}(x))^2] \tag{15}$$

$$= p_{\pi_\theta}(x)(1 - p_{\pi_\theta}(x))^2 + (1 - p_{\pi_\theta}(x))(0 - p_{\pi_\theta}(x))^2 \tag{16}$$

$$= p_{\pi_\theta}(x)(1 - p_{\pi_\theta}(x)) \tag{17}$$

which is maximized at $p_{\pi_\theta}(x) = \frac{1}{2}$. Therefore, intermediate-level prompts yield the largest expected magnitude of advantage and upper bound on gradient updates. These observations motivate a prompt curriculum strategy that prioritizes prompts with intermediate difficulty ($p_{\pi_\theta}(x) = \frac{1}{2}$) to enhance training efficiency.

# E    EXPERIMENT DETAILS

## E.1    BASELINES ALGORITHMS

We list the pseudo-code for GRPO, Pre-filter, DS, and SPEED. For the pseudo-code of GRESO, please refer to Zheng et al. (2025).

---

**Algorithm 2** GRPO

---

**Require:** Number of prompts $m$, generations per prompt $n$
1: Initialize policy $\pi_0$
2: **for** $t = 0$ to $T - 1$ **do**
3:      Sample a batch with $m$ prompts: $\mathcal{D}_m = \{x^i\}_{i=1}^m \subset \mathcal{D}$.
4:      Generate $n$ responses: $\mathcal{D}_m = \left\{(x^i, \{y^{i,j}\}_{j=1}^n)\right\}_{i=1}^m$ where $y^{i,j} \overset{\text{iid}}{\sim} \pi_t(\cdot \mid x^i)$.
5:      Update to $\pi_{t+1}$ with GRPO objective using $\mathcal{D}_m$.
6: **end for**

---

**Algorithm 3** Pre-filter

---

**Require:** Number of prompts $m$, generations per prompt $n$, pre-filter generations $n_{\text{pre}}$, thresholds $p_{\text{low}}$ and $p_{\text{high}}$
1: Initialize policy $\pi_0$
2: Generate $n_{\text{pre}}$ responses for each prompt in the dataset: $\{y^j\}_{j=1}^{n_{\text{pre}}} \sim \pi_0(\cdot|x)$ for each $x \in \mathcal{D}$
3: Filter to keep prompts with accuracy between $p_{\text{low}}$ and $p_{\text{high}}$:

$$\mathcal{D} \leftarrow \{x \in \mathcal{D} \mid p_{low} < \frac{1}{n_{\text{pre}}} \sum_{j=1}^{n_{\text{pre}}} r(x, y^j) < p_{high}\}$$

4: **for** $t = 0$ to $T - 1$ **do**
5:      Sample a batch with $m$ prompts: $\mathcal{D}_m = \{x^i\}_{i=1}^m \subset \mathcal{D}$.
6:      Generate $n$ responses: $\mathcal{D}_m = \left\{(x^i, \{y^{i,j}\}_{j=1}^n)\right\}_{i=1}^m$ where $y^{i,j} \overset{\text{iid}}{\sim} \pi_t(\cdot \mid x^i)$.
7:      Update to $\pi_{t+1}$ with GRPO objective using $\mathcal{D}_m$.
8: **end for**

---

**Algorithm 4** Dynamic-sampling (DS)

---

**Require:** Number of prompts $m$, generations per prompt $n$, sampling parameter $k$
1: Initialize policy $\pi_0$, $\mathcal{D}_{\text{buffer}} \leftarrow \varnothing$
2: **for** $t = 0$ to $T - 1$ **do**
3:      **while** $|\mathcal{D}_{\text{buffer}}| < m$ **do**
4:          Sample a batch with $km$ prompts: $\mathcal{D}_{km} = \{x^i\}_{i=1}^{km} \subset \mathcal{D}$.
5:          Generate $n$ responses: $\mathcal{D}_{km} = \left\{(x^i, \{y^{i,j}\}_{j=1}^n)\right\}_{i=1}^{km}$ where $y^{i,j} \overset{\text{iid}}{\sim} \pi_t(\cdot \mid x^i)$
6:          Select prompts with mean reward between 0 and 1:

$$\mathcal{D}_{\text{buffer}} \leftarrow \mathcal{D}_{\text{buffer}} \cup \{(x, \{y^j\}_{j=1}^n) \in \mathcal{D}_{km} \mid 0 < \frac{1}{n} \sum_{j=1}^n r(x, y^j) < 1\}$$

7:      **end while**
8:      Sample a batch with $m$ prompts: $\mathcal{D}_m = \left\{(x^i, \{y^{i,j}\}_{j=1}^n)\right\}_{i=1}^m \subset \mathcal{D}_{\text{buffer}}$.
9:      $\mathcal{D}_{\text{buffer}} \leftarrow \varnothing$
10:     Update to $\pi_{t+1}$ with GRPO objective using $\mathcal{D}_m$.
11: **end for**

---

---

**Algorithm 5** SPEED

---

**Require:** Number of prompts $m$, generations per prompt $n$, sampling parameter $k$, screening number of responses $n_{\text{init}}$ ($n \geq n_{\text{init}}$)

1: Initialize policy $\pi_0$, $\mathcal{D}_{\text{buffer}} \leftarrow \varnothing$, $\mathcal{D}_{\text{accepted}} \leftarrow \varnothing$
2: **for** $t = 0$ to $T - 1$ **do**
3:     **while** $|\mathcal{D}_{\text{buffer}}| < m$ **do**
4:         Sample a batch with $km$ prompts: $\mathcal{D}_{km} = \{x^i\}_{i=1}^{km} \subset \mathcal{D}$.
5:         Generate $n_{\text{init}}$ times for $\mathcal{D}_{km}$ and $n - n_{\text{init}}$ times for $\mathcal{D}_{\text{accepted}}$:

$$\mathcal{D}_{km} = \left\{ \left(x^i, \{y^{i,j}\}_{j=1}^{n_{\text{init}}}\right)\right\}_{i=1}^{km} \text{ where } y^{i,j} \overset{\text{iid}}{\sim} \pi_t(\cdot \mid x^i)$$

$$\mathcal{D}_{\text{accepted}} \leftarrow \mathcal{D}_{\text{accepted}} \cup \left\{ \left(x^i, \{y^{i,j}\}_{j=1}^{n-n_{\text{init}}}\right)\right\}_{i=1}^{|\mathcal{D}_{\text{accepted}}|} \text{ where } y^{i,j} \overset{\text{iid}}{\sim} \pi_t(\cdot \mid x^i).$$

6:         Add $\mathcal{D}_{\text{accepted}}$ to $\mathcal{D}_{\text{buffer}}$: $\mathcal{D}_{\text{buffer}} \leftarrow \mathcal{D}_{\text{buffer}} \cup \mathcal{D}_{\text{accepted}}$
7:         Select prompts with mean reward between 0 and 1 and add to $\mathcal{D}_{\text{accepted}}$:

$$\mathcal{D}_{\text{accepted}} \leftarrow \left\{ \left(x, \{y^j\}_{j=1}^{n_{\text{init}}}\right) \in \mathcal{D}_{km} \mid 0 < \frac{1}{n_{\text{init}}} \sum_{j=1}^{n_{\text{init}}} r(x, y^j) < 1 \right\}$$

8:     **end while**
9:     Sample a batch with $m$ prompts: $\mathcal{D}_m = \left\{ \left(x^i, \{y^{i,j}\}_{j=1}^{n}\right)\right\}_{i=1}^{m} \subset \mathcal{D}_{\text{buffer}}$.
10:     $\mathcal{D}_{\text{buffer}} \leftarrow \mathcal{D}_{\text{buffer}} \smallsetminus \mathcal{D}_m$
11:     Update to $\pi_{t+1}$ with GRPO objective using $\mathcal{D}_m$.
12: **end for**

---

### E.2 DATASET, MODEL, REWARD, EVALUATION DETAILS

We adopt the exact same setting in our preliminary investigation. Refer to Appendix B for details on datasets, models, rewards, and evaluations.

### E.3 HYPERPARAMETERS

We list the hyperparameters for each method below. We tune the sample batch size / sampling parameter for DS, SPEED, GRESO, and PCL with the **bolded** one as the best one. For PCL, we always use the same-sized model as the value model for the main results. The learning rate of the value model in PCL is tuned within $\{1e-6, 3e-6, 1e-5\}$. All learning rates make the value model converge to the same performance but larger learning rates are more unstable than the smaller one. Therefore, we pick 1e-6 as the learning rate for the value model. We also tune the sampling parameter $k$ from $\{2, 4, 8\}$ and observe that it has a minor effect on the accuracy of the value model. Larger values of $k$ lead to higher effective ratios, as the filtering becomes more aggressive. However, increasing $k$ beyond 4 yields only marginal improvements in the effective ratio. Therefore, we set $k = 4$ for all experiments on PCL. An ablation on the value model size is included in Appendix G.

| Dataset | Method | Parameters | |
|---|---|---|---|
| MATH | GRPO | $m = 512$ 
 $lr = 8e-6$ | $n = 16$ |
| DeepScaleR | GRPO | $m = 512$ 
 $lr = 4e-6$ | $n = 16$ |
| MATH | Pre-filter | $m = 512$ 
 $lr = 8e-6$ 
 $p_{\text{low}} = 0$ | $n = 16$ 
 $n_{\text{pre}} = 16$ 
 $p_{\text{high}} = 1$ |
| DeepScaleR | Pre-filter | $m = 512$ 
 $lr = 4e-6$ 
 $p_{\text{low}} = 0$ | $n = 16$ 
 $n_{\text{pre}} = 16$ 
 $p_{\text{high}} = 1$ |
| MATH | DS | $m = 512$ 
 $lr = 8e-6$ | $n = 16$ 
 $k = 1/\mathbf{2}/4$ |
| DeepScaleR | DS | $m = 512$ 
 $lr = 4e-6$ | $n = 16$ 
 $k = 1/\mathbf{2}/4$ |
| MATH | SPEED | $m = 512$ 
 $lr = 8e-6$ 
 $n_{\text{init}} = \mathbf{4}/8$ | $n = 16$ 
 $k = 1/\mathbf{2}/4$ |
| DeepScaleR | SPEED | $m = 512$ 
 $lr = 4e-6$ 
 $n_{\text{init}} = \mathbf{4}/8$ | $n = 16$ 
 $k = 1/\mathbf{2}/4$ |
| MATH | GRESO | $m = 512$ 
 $lr = 8e-6$ 
 $p_{easy} = 0.5$ 
 $\alpha_{easy} = 0.083$ 
 $\Delta p = 0.01$ | $n = 16$ 
 $B_{\text{r}}^{\text{default}} = \mathbf{768}/1024$ 
 $p_{hard} = 0.5$ 
 $\alpha_{hard} = 0.167$ |
| DeepScaleR | GRESO | $m = 512$ 
 $lr = 4e-6$ 
 $p_{easy} = 0.5$ 
 $\alpha_{easy} = 0.083$ 
 $\Delta p = 0.01$ | $n = 16$ 
 $B_{\text{r}}^{\text{default}} = \mathbf{768}/1024$ 
 $p_{hard} = 0.5$ 
 $\alpha_{hard} = 0.167$ |
| MATH | PCL | $m = 512$ 
 $lr = 8e-6$ 
 $\tau = 0.5$ | $n = 16$ 
 $lr_{\text{critic}} = \mathbf{1e\text{-}6}/3e\text{-}6/1e\text{-}5$ 
 $k = 2/\mathbf{4}/8$ |
| DeepScaleR | PCL | $m = 512$ 
 $lr = 4e-6$ 
 $\tau = 0.5$ | $n = 16$ 
 $lr_{\text{critic}} = \mathbf{1e\text{-}6}/3e\text{-}6/1e\text{-}5$ 
 $k = 2/\mathbf{4}/8$ |

## F COMPLETE EXPERIMENT RESULTS

Table 13 shows the full results on DeepScaleR. PCL consistently achieves either the highest final accuracy or substantially reduced wall-clock time at comparable accuracy. We summarizes the results on DeepScaleR below, comparing PCL with the best or the second best method for each model in terms of test accuracy:

- Qwen3-8B-Base: PCL is the top-performing method and converges in 41.8 hours, compared to 69.5 hours for DS, which also achieves lower average accuracy. PCL converges **39.8%** faster.

- Qwen3-4B-Base: PCL achieves 45.7 test accuracy and converges in 32.8 hours, compared to 45.8 accuracy and 40.1 hours for DS. PCL therefore converges **18.2%** faster while maintaining similar performance.

- Qwen3-1.7B-Base: PCL is again the best-performing method and converges in 23.3 hours, whereas Pre-filter requires 44.2 hours and yields lower average accuracy. PCL converges **47.3%** faster.

- Llama3.2-3B-it: PCL reaches 26.5 accuracy in 28.7 hours, compared to 26.7 accuracy in 40.6 hours for DS, achieving similar accuracy with **29.3%** faster convergence.

The consistent 1.3 to 2× reductions in convergence time are practically impactful in real RL pipelines where rollouts are the dominant cost.

Table 13: **Full Results on DeepScaleR.** For each metric, the best-performing method is highlighted in **bold**, and the second-best is underlined. Time is the sum of training and generation time of the checkpoint that achieves the best average performance (excluding validation / checkpointing) in hours.

| DeepScaleR | Method | MATH500 | Olymp. | Minerva Avg@4 | AMC23 Avg@32 | AIME24 Avg@32 | AIME25 Avg@32 | Avg. | Time |
|---|---|---|---|---|---|---|---|---|---|
| Qwen3-8B-Base | $\pi_{\text{ref}}$ | 70.2 | 34.3 | 29.8 | 49.1 | 15.8 | 8.8 | 34.7 | / |
| | GRPO | 87.2 | 57.9 | 45.3 | 70.1 | 25.3 | 22.7 | 51.4 | 43.0 |
| | Pre-filter | 86.4 | 54.6 | 44.2 | 69.8 | 26.9 | 22.6 | 50.7 | 67.4 |
| | DS | 87.2 | 55.3 | 45.7 | 71.5 | 24.9 | 24.2 | 51.5 | 69.5 |
| | SPEED | 82.4 | 46.4 | 40.3 | 66.6 | 21.1 | 15.7 | 45.5 | **19.3** |
| | PCL | 88.4 | 56.2 | 46.8 | 71.2 | 25.2 | 23.9 | **52.0** | 41.8 |
| Qwen3-4B-Base | $\pi_{\text{ref}}$ | 65.8 | 34.4 | 26.9 | 47.3 | 10.9 | 7.1 | 32.1 | / |
| | GRPO | 83.4 | 51.0 | 40.1 | 60.7 | 16.1 | 20.7 | 45.3 | 45.5 |
| | Pre-filter | 83.4 | 47.8 | 40.0 | 60.2 | 18.8 | 16.2 | 44.4 | 39.0 |
| | DS | 83.2 | 51.6 | 41.2 | 62.4 | 18.5 | 18.0 | **45.8** | 40.1 |
| | SPEED | 79.4 | 45.4 | 38.3 | 60.3 | 15.7 | 14.5 | 42.3 | **10.7** |
| | PCL | 83.0 | 50.6 | 40.9 | 60.8 | 19.4 | 19.4 | 45.7 | 32.8 |
| Qwen3-1.7B-Base | $\pi_{\text{ref}}$ | 57.0 | 23.9 | 21.8 | 29.0 | 3.8 | 1.1 | 22.8 | / |
| | GRPO | 72.4 | 37.7 | 31.2 | 44.9 | 11.2 | 6.7 | 34.0 | 46.2 |
| | Pre-filter | 74.0 | 36.5 | 32.6 | 45.6 | 11.7 | 7.8 | 34.7 | 44.2 |
| | DS | 73.2 | 36.9 | 31.9 | 42.7 | 10.8 | 7.7 | 33.9 | 41.7 |
| | SPEED | 73.0 | 34.4 | 30.2 | 37.2 | 9.2 | 7.1 | 31.8 | **22.7** |
| | PCL | 74.4 | 35.6 | 31.5 | 46.3 | 12.5 | 9.2 | **34.9** | 23.3 |
| Llama3.2-3B-it | $\pi_{\text{ref}}$ | 42.8 | 12.3 | 13.8 | 19.7 | 4.6 | 0.4 | 15.6 | / |
| | GRPO | 55.2 | 23.1 | 22.6 | 40.0 | 13.3 | 0.0 | 25.7 | 47.5 |
| | Pre-filter | 56.8 | 24.5 | 23.3 | 35.5 | 16.5 | 0.7 | 26.2 | 44.8 |
| | DS | 57.2 | 23.3 | 24.1 | 37.1 | 17.5 | 1.0 | **26.7** | 40.6 |
| | SPEED | 51.4 | 20.2 | 20.1 | 32.0 | 10.6 | 0.8 | 22.5 | **3.86** |
| | PCL | 58.8 | 23.9 | 24.0 | 35.2 | 15.0 | 2.1 | 26.5 | 28.7 |

# G    VALUE MODEL SIZE ABLATION

We ablate the value model used in PCL across three different model sizes, with results on explained variance of the value prediction shown in Figure 18. Similar to Section 5, we use the average reward of 16 generations for each prompt as the ground-truth $p(x)$. Overall, larger value models exhibit faster convergence compared to smaller ones. On the MATH dataset, all three value models eventually converge to similar performance levels. However, on the larger DeepScaleR dataset, we observe a substantial gap after 100 training steps: the smaller value model significantly underperforms relative to its larger counterparts.

We hypothesize that the smaller value model may require more training steps to reach comparable accuracy and that, given sufficient time, all models could eventually converge to a similar point. Nonetheless, this result highlights the benefit of larger value models in the early stages of training, especially on large-scale datasets. In addition, 4B value model performs similarly to a 8B model. This suggests that the size of the value model does not need to scale proportionally with the policy. While we have not tested larger policies due to resource constraints, we expect diminishing returns from larger value models.

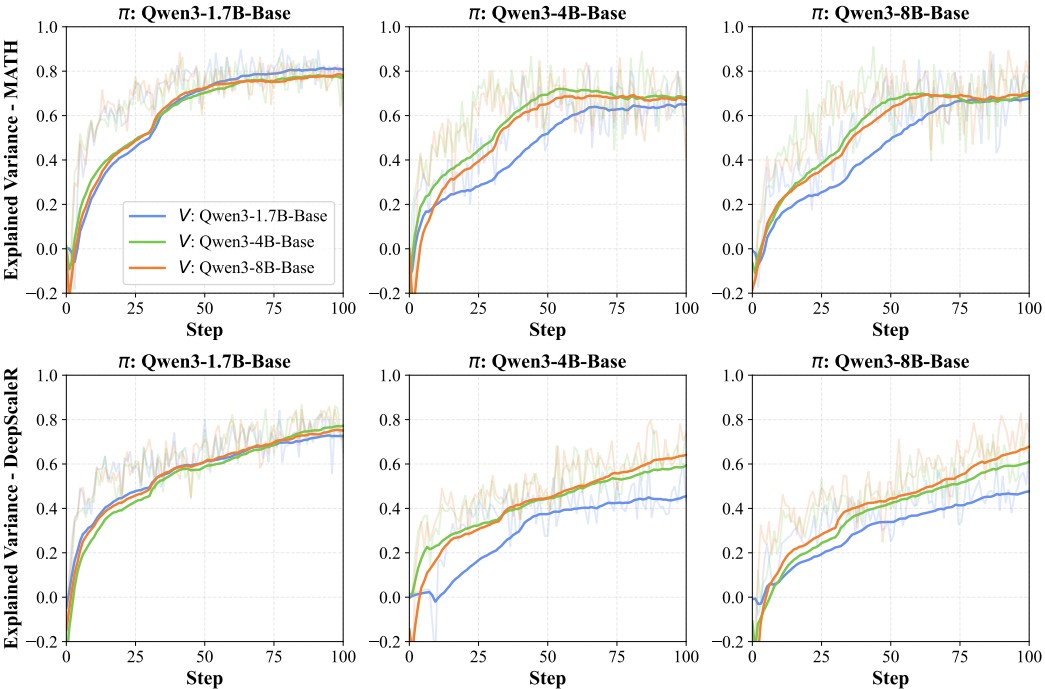

Figure 18: Explained variance on MATH and DeepScaleR with different combinations of Qwen3 base models (1.7B / 4B / 8B) for policy ($\pi$) and the value model ($V$).

## H  STOCHASTIC BATCH SELECTION

In this section, we provide an ablation on using stochastic batch selection instead of greedy selection around threshold $\tau$. Since the reward is binary, we sample from a Beta distribution whose support naturally lies in $[0, 1]$. We set $\alpha = \beta$ for the Beta distribution and ablate $\alpha$ and $\beta$ from 2 to 10, higher values indicate more focus on 0.5 difficulty prompts. From our experiments, greedy sampling and sampling based on Beta distribution perform similarly when $\alpha$ and $\beta$ are large. However, using smaller values of $\alpha$ and $\beta$, which spread the probability mass more broadly, actually degrades convergence, likely because it focuses less on 0.5 difficulty prompts. The results are detailed below for Llama3.2-3B-it on MATH dataset.

Table 14: Results on MATH with Llama3.2-3B-it under different $(\alpha, \beta)$ settings. Time is the sum of training and generation time of the checkpoint that achieves the best average performance (excluding validation/checkpointing) in hours.

| $\alpha$ | $\beta$ | MATH500 | Time |
|---|---|---|---|
| 2 | 2 | 56.4 | **10.8** |
| 5 | 5 | 56.2 | 15.9 |
| 10 | 10 | 57.2 | 15.6 |
| greedy | | **57.8** | 14.3 |

We therefore adopt greedy top-$m$ selection based on threshold $\tau$, which is simple without additional hyperparameters, and already delivers strong performance.

## I  TRAINING TIME BREAKDOWN

Table 15: Breakdown of per-step policy and value training/inference time in seconds, averaged over the full training run, when training Qwen3-8B-Base on MATH with three different sized value models (Qwen3-8B-Base, Qwen3-4B-Base, Qwen3-1.7B-Base).

| Value Model | Generation | Value Computation | Policy Update | Value Update |
|---|---|---|---|---|
| Qwen3-8B-Base | 573 | **2.99** | 769 | **20.5** |
| Qwen3-4B-Base | 567 | **2.11** | 763 | **12.4** |
| Qwen3-1.7B-Base | 582 | **1.16** | 760 | **8.03** |

For reasoning tasks, the prompt is typically much shorter than the response, which significantly reduces value model compute and memory compared to policy updates. We include a detailed breakdown of the time per step when training Qwen3-8B-Base on MATH with three different sized value models (Qwen3-8B-Base, Qwen3-4B-Base, Qwen3-1.7B-Base). The results are shown in Table 15. Value computation time takes less than 1% of the generation time, and value model update time takes less than 3% of the time to update the policy.

## J LIMITATIONS

While PCL demonstrates strong empirical performance across a range of models and datasets, our study has several limitations that open avenues for future work.

**Purely on-policy setting.** Our experiments are conducted entirely in a purely on-policy RL setting, where new generations are sampled after each policy update. While this simplifies the analysis and avoids additional hyperparameters (e.g., clipping), it may reduce the generalization to more complex training pipelines that leverage off-policy data or replay buffers.

**Focus on synchronous setting.** Our preliminary investigation and PCL are evaluated in a synchronous training setup where data generation and policy updates are alternated step-by-step. However, many large-scale RL pipelines for LLMs adopt asynchronous architectures for better throughput (Wu et al., 2025; Fu et al., 2025). Extending our analysis and PCL to asynchronous settings may require more sophisticated value model training and prompt selection strategies to handle stale or partially updated policies.

**Relatively short context lengths.** We limit our experiments to a maximum context length of 4,096 tokens due to compute constraints. While this setting is sufficient for the datasets used (e.g., MATH, DeepScaleR), real-world LLM deployments often involve much longer contexts. From our analysis, for longer context length, the batch size that transitions from sub-linear to linear generation time is larger. Future work could explore the interplay between prompt difficulty, batch decomposition, and context length in long-context regimes.

**Limited training horizon.** Our experiments are constrained to relatively short training runs (e.g., 2–3 days), which may not fully capture long-term convergence behavior, especially for larger models and datasets. Although we observe strong early-stage performance, it remains an open question whether our analysis in Section 3 would generalize to much longer training runs.

**Limited task diversity.** Our current evaluation is restricted to mathematical reasoning benchmarks, which provide a clean and controlled environment for studying curriculum mechanisms but do not capture the full breadth of real-world LLM applications. In particular, extending PCL to domains such as code generation would require compiling and executing test cases during reward computation, substantially increasing runtime and compute cost. Due to limited computational resources, we were unable to include such experiments. We view applying PCL to a broader set of tasks, including coding, multi-step tool use, and open-ended dialogue, as an important direction for future work that would further validate and potentially refine the proposed curriculum framework.

