# OpenReview forum: "Prompt Curriculum Learning for Efficient LLM Post-Training"
_ICLR.cc/2026/Conference — ICLR 2026 Poster_

### Official Review · Reviewer_BGex · 2025-11-01

**Soundness:** 3
**Presentation:** 3
**Contribution:** 3
**Rating:** 6
**Confidence:** 3

**Summary:**

The paper investigates how batching and prompt selection affect reinforcement learning post-training of large language models, identifying two key empirical findings: an optimal total batch size where rollout time scaling transitions from sublinear to linear, and the highest learning efficiency when prompts have intermediate difficulty (success probability ≈ 0.5). Building on these insights, the authors propose Prompt Curriculum Learning (PCL), which trains a lightweight value model to estimate per-prompt reward and selects prompts near a target difficulty without costly on-policy rollouts. Across several math reasoning benchmarks (MATH, Olympiad-Bench, AIME, etc.) and models (Qwen3, Llama3.2), PCL matches or surpasses other post-training baselines.

**Strengths:**

1. The paper focuses on a practically important yet underexplored question, how batching and prompt difficulty affect the efficiency of RL post-training for large language models, and grounds its approach in two clear empirical observations.

2. Through systematic experiments, the study identifies an optimal total batch size and shows that prompts of intermediate difficulty yield the most effective learning signal, offering useful insights for designing efficient RL fine-tuning pipelines.

3. The method is validated across several models and math reasoning benchmarks with comprehensive ablations, demonstrating consistent performance trends and providing a transparent analysis of efficiency, stability, and cost trade-offs.

**Weaknesses:**

1. The effectiveness of PCL may heavily rely on the accuracy of the value model. In more complex settings, a small or undertrained value model may fail to provide reliable estimates, leading to suboptimal prompt selection and degraded performance.

2. Concentrating too much on p(x)≈0.5 (especially with large n, small m) lowers prompt diversity, which can reduce robustness and even degrade accuracy beyond some point.

3. The experiments focus almost entirely on math reasoning tasks, leaving it unclear whether the findings on optimal batch size and prompt difficulty generalize to more diverse domains such as code generation or instruction following.

4. If the average reward over the dataset is far from 0.5 (label imbalance), filtering near 0.5 may implicitly rebalance, which can be good or bad depending on goals; it’s another place PCL’s assumptions can break.

**Questions:**

Please refer to the weaknesses section.

---

> ### Author Response · Authors · 2025-11-22
>
> We thank the reviewer for the thoughtful and constructive feedback.
>
> > Weaknesses 1
>
> We agree that the reliability of the value model is a critical factor for PCL’s performance. To empirically assess this, we include an ablation study in Appendix G, which shows that a 4B value model performs comparably to an 8B value model when paired with a Qwen3-8B-Base policy. This suggests that the value model does not need to scale proportionally with the policy to provide effective prompt selection, and relatively small value models can produce sufficiently accurate difficulty estimates to guide curriculum learning.
>
> In addition, PCL does not require precise value estimates, and only needs a coarse relative ranking near a threshold. We include experiments using stochastic sampling below. Since our reward is binary, we sample from a Beta distribution whose support naturally lies in [0, 1]. We set $\alpha=\beta$ and ablate $\alpha$ and $\beta$ from 2 to 10, higher values indicate more focus on 0.5 difficulty prompts. From our experiments, greedy sampling and sampling based on Beta distribution perform similarly when $\alpha$ and $\beta$ are large. The results are detailed below for Llama3.2-3B-it on MATH dataset:
>
> | $\alpha$ | $\beta$ | MATH500 | Time to Convergence (hr) |
> |-------------|------------|---------------|--------|
> |       2       |      2      |  56.4         | 10.8 |
> | 5             | 5           |  56.2         | 15.9 |
> | 10           | 10         |  57.2         | 15.6 |
> | greedy   | greedy   |  **57.8**   | 14.3 |
>
> When the Beta distribution has high $\alpha$ and $\beta$, the performance is similar to greedy sampling, showing that PCL is robust to noise.
>
> Furthermore, PCL is robust to value model imperfections. In the worst case, if the value model provides purely random scores, PCL reduces to standard GRPO training (i.e., uniform sampling over prompts), which remains a valid baseline. This robustness makes PCL a safe enhancement: while an accurate value model improves convergence, a poorly trained one does not catastrophically degrade performance.
>
> > Weaknesses 2
>
> We note that PCL has a tunable parameter $k$ that controls the strength of filtering: decreasing $k$ broadens the sampled difficulty range, and increasing $k$ sharpens the focus around the target threshold. This provides a mechanism to adjust the curriculum if the dataset exhibits extreme distributions of difficulty or the assumption on the prompt-level generalization does not hold.
>
> In our experiments, we set $k=4$ and found it to be effective across all models and datasets. During parameter sweep, we observe that both weaker filtering ($k=2$) and overly aggressive filtering ($k=8$) lead to a slight degraded convergence and lower final performance, supporting the reviewer’s intuition that diversity–efficiency trade-offs exist. We clarify this observation and include a brief discussion in line 1847 - 1851.
>
> > Weaknesses 3
>
> We agree that testing PCL on more diverse reasoning tasks is an important direction for future work. Coding tasks, however, require compilation and execution of test cases during reward computation, which substantially increases runtime and compute cost. Due to limited resources, we were unable to include such experiments. We added this clarification to the Limitations section (line 2077 - 2084).
>
> For instruction following tasks, the rewards are typically continuous. We suspect that focusing on intermediate difficulty prompts would matter less for continuous rewards as the effective ratio is rarely zero (i.e. all responses have the exact same reward). However, systematically understanding the role of prompt difficulty is valuable future work.
>
> > Weaknesses 4
>
> In the extreme case where all prompts are initially very difficult, selecting intermediate prompts simply corresponds to choosing the relatively easier subset among them, still providing a useful learning signal. Conversely, if all prompts are initially very easy, the “intermediate” region would correspond to relatively harder prompts, again ensuring training progression. Thus, PCL automatically adapts to the empirical difficulty distribution: it always selects the most informative region of the prompt space, even if that region shifts dynamically over time.

---

### Official Review · Reviewer_SyDP · 2025-11-02

**Soundness:** 3
**Presentation:** 3
**Contribution:** 3
**Rating:** 6
**Confidence:** 3

**Summary:**

This paper investigates how prompt difficulty and batch configuration affect the efficiency of RL post-training for LLMs. The authors show empirically that there exists an optimal batch size that balances generation time and gradient quality, and that prompts of intermediate difficulty (where the model has roughly a 50% chance of success) are the most sample-efficient for model convergence. They propose Prompt Curriculum Learning (PCL), which trains a value model V(x) to estimate expected reward and select prompts closest to a target threshold $\tau = 0.5$, enabling efficient on-policy filtering without expensive rollouts. PCL is shown to either exceed the performance of baseline methods or require less time to train for a similar performance.

**Strengths:**

* Advances the efficiency of LLM post-training based on interesting empirical findings about the batch size and the intermediate difficulty of prompts, which are supported by broad ablations across models, datasets, and a significant spend of resources.
* The value network to select prompts is lightweight and easy to adopt.
* The experimental section is well presented and results show the method reaches highest accuracy, or reaches baseline accuracy significantly faster.

**Weaknesses:**

* The method assumes that the dataset always contains a sufficient number of prompts with approximately 50% success probability. However, during early training, there may be very few intermediate-difficulty prompts, which could hinder learning and cause training to stagnate, as the model would not encounter harder examples. Similarly, toward the end of training, if the model improves rapidly, only a small subset of prompts may continue to be selected, potentially biasing the training distribution. The paper does not appear to analyze or address these edge cases, which could affect the robustness of the proposed curriculum.
* Filtering may introduce distributional bias by over-representing specific mathematical subtypes while under-exposing the model to both foundational easy patterns and extremely difficult reasoning structures. This could lead to topic drift or a collapse toward narrower reasoning styles, as the curriculum disproportionately focuses on intermediate-difficulty prompts. The paper does not provide an analysis of how filtering affects topic coverage or reasoning diversity over training.
* The lack of variance reporting weakens the paper's claims of robustness, as the absence of standard deviation metrics makes it difficult to assess the statistical significance of the reported improvements. Moreover, the paper does not present results across multiple random seeds for the final evaluations, leaving open the possibility that the observed gains may be sensitive to initialization or sampling noise.

**Questions:**

1. What is the empirical distribution of prompt difficulty $\pi(x)$ across the full dataset for the initial policy and throughout training? Without this, it is unclear whether intermediate-difficulty prompts are abundant, rare, or emergent as a function of policy improvement.
2. If intermediate prompts are scarce early on, is learning bottlenecked?
3. Did you evaluate selecting a band of difficulties (e.g., $\tau \pm \epsilon$) such as 0.2–0.8 or 0.4–0.6, instead of a point target? I think this may alleviate the scarcity issue by increasing sample diversity while still maintaining meaningful gradient feedback.

---

> ### Author Response · Authors · 2025-11-22
>
> We thank the reviewer for the thoughtful and constructive feedback.
>
> > The method assumes that the dataset always contains a sufficient number of prompts with approximately 50% success probability. However, during early training, there may be very few intermediate-difficulty prompts, which could hinder learning and cause training to stagnate, as the model would not encounter harder examples. Similarly, toward the end of training, if the model improves rapidly, only a small subset of prompts may continue to be selected, potentially biasing the training distribution. The paper does not appear to analyze or address these edge cases, which could affect the robustness of the proposed curriculum.
>
> We appreciate the reviewer’s insightful question. We clarify that prompt filtering methods, including ours, implicitly rely on prompt-level generalization, i.e., improving on a selected subset of prompts also improves performance on the unselected ones (line 534). This assumption underlies many curriculum learning methods.
>
> During the early stage, when the model is weak, most prompts will indeed appear difficult. In this case, “intermediate difficulty” simply corresponds to the easier subset of these difficult prompts. As training progresses, the distribution of prompt difficulty shifts, and previously hard examples begin to enter the intermediate region. Thus, PCL naturally transitions from easier to harder examples over time without requiring any explicit schedule. This is discussed in line 442.
>
> Similarly, near convergence, only a small fraction of prompts remain unsolved. PCL will then selectively focus on these truly hard examples, which is desirable because they represent the remaining learning signal. In both early and late stages, targeting the ~50% difficulty region serves as a principled way to adaptively select informative prompts over training.
>
> We also note that PCL has a tunable parameter $k$ that controls the strength of filtering: decreasing $k$ broadens the sampled difficulty range, and increasing $k$ sharpens the focus around the target threshold. This provides a mechanism to adjust the curriculum if the dataset exhibits extreme distributions of difficulty or the assumption on the prompt-level generalization does not hold.
>
> > Filtering may introduce distributional bias by over-representing specific mathematical subtypes while under-exposing the model to both foundational easy patterns and extremely difficult reasoning structures. This could lead to topic drift or a collapse toward narrower reasoning styles, as the curriculum disproportionately focuses on intermediate-difficulty prompts. The paper does not provide an analysis of how filtering affects topic coverage or reasoning diversity over training.
>
> Measuring reasoning diversity and topic coverage in open-ended mathematical reasoning is non-trivial. The goal of the paper focuses on improving convergence, specifically, maximizing performance under fixed compute and wall-clock budgets. We empirically demonstrate that PCL could improve convergence over existing baselines on a variety of datasets and benchmarks, suggesting that the curriculum does not harm and could enhance generalization to diverse reasoning tasks. Measuring reasoning diversity and topic coverage falls outside the primary scope of this work and a systematic study of how filtering affects conceptual and topical coverage is an interesting direction for future work.

---

> ### Author Response · Authors · 2025-11-22
>
> > Did you evaluate selecting a band of difficulties (e.g., $\pi \pm \epsilon$) such as 0.2–0.8 or 0.4–0.6, instead of a point target? I think this may alleviate the scarcity issue by increasing sample diversity while still maintaining meaningful gradient feedback.
>
> During the development of PCL, we experiment with stochastic batch selection. Since our reward is binary, we sample from a Beta distribution whose support naturally lies in [0, 1]. We set $\alpha=\beta$ and ablate $\alpha$ and $\beta$ from 2 to 10, higher values indicate more focus on 0.5 difficulty prompts. From our experiments, greedy sampling and sampling based on Beta distribution perform similarly when $\alpha$ and $\beta$ are large. However, using smaller values of $\alpha$ and $\beta$, which spread the probability mass more broadly, actually degrades convergence, likely because it focuses less on 0.5 difficulty prompts. The results are detailed below for Llama3.2-3B-it on MATH dataset:
>
> | $\alpha$ | $\beta$ | MATH500 | Time to Convergence (hr) |
> |-------------|------------|---------------|--------|
> |       2       |      2      |  56.4         | 10.8 |
> | 5             | 5           |  56.2         | 15.9 |
> | 10           | 10         |  57.2         | 15.6 |
> | greedy   | greedy   |  **57.8**   | 14.3 |
>
> We deliberately chose greedy top-m selection because it is simple and robust to implement in large-scale RL pipelines, and already yields strong accuracy–time trade-offs compared to established baselines (GRPO, DS, SPEED, GRESO). We added a discussion of this result in the revised version (line 2000 - 2020).
>
> > The lack of variance reporting weakens the paper's claims of robustness, as the absence of standard deviation metrics makes it difficult to assess the statistical significance of the reported improvements. Moreover, the paper does not present results across multiple random seeds for the final evaluations, leaving open the possibility that the observed gains may be sensitive to initialization or sampling noise.
>
> We appreciate the opportunity to clarify. While we did not include variance in the current submission, all experiments were run with fixed seeds, making our results exactly reproducible. Furthermore, in large-scale RL for LLMs, empirical results tend to be stable across runs due to the high sample count per iteration, often involving hundreds of prompt-response pairs per update and training over thousands of steps. To further reduce variance, we report results on large-scale datasets and, for smaller benchmarks such as Minerva, AMC 23 and AIME 24/25, we compute the average performance over 4 or 32 generations per prompt. This setup ensures statistical reliability even in the absence of formal error bars. We are confident that the observed claims and performance improvements are robust and not due to chance, as they are consistent across models, datasets, and evaluation metrics.
>
> > What is the empirical distribution of prompt difficulty $\pi(x)$ across the full dataset for the initial policy and throughout training? Without this, it is unclear whether intermediate-difficulty prompts are abundant, rare, or emergent as a function of policy improvement.
>
> We appreciate the reviewer’s question. In practice, while evaluating every checkpoint over the entire training dataset during training is infeasible due to large computation, we suspect that intermediate-difficulty prompts are abundant across training. Evidence for this can be seen in the middle column of Figure 16, where filtering at different thresholds (from 0 to 1) consistently yields training rewards that closely match the target threshold throughout training. This indicates that prompts span a sufficiently wide range of difficulties, and sampling around the 0.5 region is reliable in practice as the training reward after filtering is very close to 0.5.
>
> > If intermediate prompts are scarce early on, is learning bottlenecked?
>
> No. In the extreme case where all prompts are initially very difficult, selecting intermediate prompts simply corresponds to choosing the relatively easier subset among them, still providing a useful learning signal. Conversely, if all prompts are initially very easy, the “intermediate” region would correspond to relatively harder prompts, again ensuring training progression. Thus, PCL automatically adapts to the empirical difficulty distribution: it always selects the most informative region of the prompt space, even if that region shifts dynamically over time.

---

### Official Review · Reviewer_ZNyN · 2025-11-02

**Soundness:** 3
**Presentation:** 3
**Contribution:** 3
**Rating:** 6
**Confidence:** 4

**Summary:**

This paper investigates strategies for efficient post-training of large language models (LLMs) using reinforcement learning (RL). The authors first conduct a systematic study on the effects of batching and prompt difficulty on RL convergence. They identify two key findings: (1) an optimal batch size exists at the transition point from sublinear to linear growth in generation time, and (2) prompts of intermediate difficulty, where the model has approximately a 50% success rate, are the most sample-efficient for training. Based on these insights, the paper proposes Prompt Curriculum Learning (PCL), a lightweight algorithm that uses a value model to efficiently select intermediate-difficulty prompts. This avoids the high computational cost of rollout-based filtering. Experiments on mathematical reasoning benchmarks (MATH and DeepScaleR) show that PCL achieves competitive performance with significantly less training time compared to several baselines.

**Strengths:**

1. To my knowledge, this is the first work to systematically analyze the optimal batch size and the optimal decomposition into the number of prompts and generations per prompt.
2. Using an additional value model for difficulty estimation before rollout is an interesting idea.
3. The paper is well-written and easy to understand.
4. The authors have conducted comprehensive empirical evaluation to support the findings on Math datasets.

**Weaknesses:**

1. The main weakness I see is that in many cases, the performance gaps between PCL and GRPO is less than 1%, thus the significance of conducting active curriculum learning is somewhat limited in these settings. And the experimental evaluation is too narrow, as it relies solely on mathematical datasets. I suggest trying to highlight the effectiveness and test the generalization of PCL in other reasoning tasks, such as coding.
2. Some related baselines are missing for discussion and comparison (e.g. MoPPS [1]). I mean this can be added to show iteration numbers as an efficiency metrics as the ablation to show connections (Unnecessary to implement for all experiments considering the time constraint in rebuttal).
3. This paper lacks systematic theoretical analysis regarding why an additional value model is sufficient to predict prompt difficulty. (In Line 300, it says: "Note that the value model $V$ in our algorithm is one step behind the policy $\pi$, which is acceptable since each update is small with $\pi_{t+1}\approx\pi_{t}$".) However, I think this can be resolved by relating to some existing works or adding some existing theoretical findings. For example, this method belongs to a model predictive sampling strategy [2], which utilizes the optimization history to build up a risk-ware module. Note that using iteration $\pi_{t}$'s generalization to approximate $\pi_{t+1}$'s task difficulty evaluation outcome first occurs in MPTS [2], and this provides a rigorous predictive foundation. Adding reference and theoretical discussions on MPTS as this work's predictive foundation can well support the validation of the prediction results.
4. "we find that both training and inference of the value model incur negligible cost and can be
  completed under 30 seconds for each step", I think providing a detailed breakdown of time consumption for the whole training process (e.g., rollout, log_prob, policy model update, value model update, ...) would better support this claim.

[1] Can prompt difficulty be online predicted for accelerating RL finetuning of reasoning models?

[2] Model Predictive Task Sampling for Efficient and Robust Adaptation.

Overall, I am quite positive on this work in terms of novelty and contributions. I'll update the score if the above weaknesses are addressed in the revised version.

**Questions:**

1. The idea of using a value model for estimating sample difficulty is somewhat connected to LLM-as-judge methods; can you add some discussions in related work?

---

> ### Author Response · Authors · 2025-11-22
>
> We thank the reviewer for the thoughtful and constructive feedback.
>
> > The main weakness I see is that in many cases, the performance gaps between PCL and GRPO is less than 1%, thus the significance of conducting active curriculum learning is somewhat limited in these settings. And the experimental evaluation is too narrow, as it relies solely on mathematical datasets. I suggest trying to highlight the effectiveness and test the generalization of PCL in other reasoning tasks, such as coding.
>
> We agree that raw accuracy differences alone can look small in some settings, and we appreciate the opportunity to clarify the joint accuracy–time gains, which we view as the main empirical contribution. As shown in Table 1 and Appendix F, PCL consistently achieves either (i) the highest final accuracy or (ii) substantially reduced wall-clock time at comparable accuracy. On MATH, PCL achieves the best test accuracy across all model sizes and families. On DeepScaleR, PCL either achieves the best average accuracy or converges significantly faster. We summarizes the results on DeepScaleR below from Appendix F, comparing PCL with the best or the second best method for each model in terms of test accuracy:
> - Qwen3-8B-Base: PCL is the top-performing method and converges in 41.8 hours, compared to 69.5 hours for DS, which also achieves lower average accuracy. PCL converges **39.8%** faster.
> - Qwen3-4B-Base: PCL achieves 45.7 test accuracy and converges in 32.8 hours, compared to 45.8 accuracy and 40.1 hours for DS. PCL therefore converges **18.2%** faster while maintaining similar performance.
> - Qwen3-1.7B-Base: PCL is again the best-performing method and converges in 23.3 hours, whereas Pre-filter requires 44.2 hours and yields lower average accuracy. PCL converges **47.3%** faster.
> - Llama3.2-3B-it: PCL reaches 26.5 accuracy in 28.7 hours, compared to 26.7 accuracy in 40.6 hours for DS, achieving similar accuracy with **29.3%** faster convergence.
>
> The consistent **1.3 to 2$\times$** reductions in convergence time are practically impactful in real RL pipelines where rollouts are the dominant cost. To ensure this contribution is clearer, we updated the paper to emphasize the efficiency perspective (line 350 - 352, 1892 - 1910).
>
> Regarding generalization beyond mathematical datasets: we agree that testing PCL on more diverse reasoning tasks is an important direction for future work. Coding tasks, however, require compilation and execution of test cases during reward computation, which substantially increases runtime and compute cost. Due to limited resources, we were unable to include such experiments in this version. We added this clarification to the Limitations section (line 2077 - 2083). We also note that evaluating PCL in a single, well-controlled reasoning domain (mathematics) allows us to more cleanly isolate the behavior of the curriculum mechanism itself, without confounding factors coming from noisy rewards. We view broad task integration as a promising future research direction that builds on the methodological contributions established in this work.
>
> > Some related baselines are missing for discussion and comparison (e.g. MoPPS [1]). I mean this can be added to show iteration numbers as an efficiency metrics as the ablation to show connections (Unnecessary to implement for all experiments considering the time constraint in rebuttal).
>
> We thank the reviewer for highlighting this related work. We included MoPPS in the Related Work section and discuss its connection to our method in the revision (line 487 - 489).
>
> We would like to emphasize that our baseline coverage already includes several recent state-of-the-art prompt filtering methods, Dynamic Sampling (March 2025), SPEED (June 2025), and GRESO (June 2025), which we believe provides a comprehensive comparison against the current literature.
>
> MoPPS is conceptually similar to dictionary-style methods such as GRESO, in that it maintains a Beta posterior for each prompt. As noted in line 320, we are limited to a single epoch over DeepScaleR due to its large scale, which is the reason we were unable to evaluate GRESO under that setting. The same constraint applies to MoPPS, which also updates the posterior every epoch, making it infeasible to run for only one epoch.

---

> ### Author Response · Authors · 2025-11-22
>
> > This paper lacks systematic theoretical analysis regarding why an additional value model is sufficient to predict prompt difficulty…
>
> We thank the reviewer for this excellent suggestion and for pointing us to MPTS [2]. We incorporated a discussion of this work in the revised version (line 524 - 526).
>
> > "we find that both training and inference of the value model incur negligible cost and can be completed under 30 seconds for each step", I think providing a detailed breakdown of time consumption for the whole training process (e.g., rollout, log_prob, policy model update, value model update, ...) would better support this claim.
>
> We thank the reviewer for this helpful suggestion. As shown below, we include a detailed breakdown of the time per step when training Qwen3-8B-Base on MATH with three different sized value models (Qwen3-8B-Base, Qwen3-4B-Base, Qwen3-1.7B-Base). Value-related computation accounts for a small fraction of total time:
>
> | Policy | Value Model | Generation Time (s) | Value Computation Time (s) | Policy Update Time (s) | Value Model Update Time (s) |
> |-------------|------------|---------|--------|--------|--------|
> | Qwen3-8B-Base | Qwen3-8B-Base | 573 | **2.99** | 769 | **20.5** |
> | Qwen3-8B-Base | Qwen3-4B-Base | 567 | **2.11** | 763 | **12.4** |
> | Qwen3-8B-Base | Qwen3-1.7B-Base | 582 | **1.16** | 760 | **8.03** |
>
> Value computation takes less than 1% of the generation time, and value model update time takes less than 3% of the time to update the policy.
>
> We again thank the reviewer for the insightful comments. We incorporated the clarifications and additional discussion suggested above in the final revision (line 2022 - 2040).
>
> References:
>
> [1] Can prompt difficulty be online predicted for accelerating RL finetuning of reasoning models?
>
> [2] Model Predictive Task Sampling for Efficient and Robust Adaptation.

---

### Official Review · Reviewer_MXSJ · 2025-11-03

**Soundness:** 4
**Presentation:** 4
**Contribution:** 3
**Rating:** 8
**Confidence:** 4

**Summary:**

This paper systematically studies how batch size and prompt difficulty affect LLM post-training dynamics and produces two key findings. First, the optimal batch size occurs at the transition point between sub-linear and linear generation time scaling. Second, they empirically confirm that prompts of intermediate difficulty (where the model has a ~50% success rate) provide the highest gradient norms and lead to the best test accuracy, validating insights from prior work.

Building on these findings, the paper proposes Prompt Curriculum Learning (PCL). PCL trains a value model alongside the main policy to estimate a prompt's difficulty (expected reward) with a single forward pass. At each step, it greedily samples prompts whose predicted value is closest to 0.5. This value model serves as a lightweight and on-policy substitute for computationally expensive methods like rollout-based filtering and avoids the off-policy staleness of dictionary-based approaches, proving to be ~12-17 times faster at identifying informative prompts.

Empirical results across various math reasoning benchmarks show that PCL either achieves state-of-the-art performance or reaches comparable results in significantly less training time.

**Strengths:**

- The paper contains systematic and rigorous experimentation to empirically validate all its core claims on optimal batch size, prompt difficulty and the effectiveness of PCL.
 - PCL is a relatively lightweight algorithm that addresses the high computational cost of rollout-based filtering and the off-policy staleness of dictionary-based methods.
 - The value model trained has reasonable accuracy (same as 3 samples per prompt) and provides a clear enough signal on prompt difficulty and significantly reduces training time during the filtering stage.
 - The paper is well presented and easy to follow with a clear narrative.

**Weaknesses:**

- A core assumption in the paper is that rewards are binary. There are many tasks like creative writing where partial rewards are the norm. It is not clear how intermediate difficulty can be mathematically determined in those cases. Similarly, it is unclear whether the value model would succeed in accurately estimating the value of a prompt in such such tasks.
 - For practical reasons, oftentimes rollouts and policy updates happen simultaneously. In such a setup the value model would be even further behind the policy and it is not clear if the accuracy of value prediction would be affected in any way.
 - Filtering exclusively for intermediate difficult could lead to the model regressing on easy tasks and never improving on the hard ones. An interesting experiment could be to do weighted sampling of the prompts during training instead of greedy sampling to allow some easy and hard prompts to also be included in training.
 - While more efficient than rollouts, PCL requires training and maintaining a second, often same-sized, value model, which doubles the memory footprint compared to single-model approaches.

**Questions:**

Please see weaknesses

---

> ### Author Response · Authors · 2025-11-22
>
> We thank the reviewer for the thoughtful and constructive feedback.
>
> > Weaknesses 1
>
> We agree that extending PCL beyond binary rewards is an important direction. We discuss the potential extension in line 520. We suspect that focusing on intermediate difficulty prompts would matter less for continuous rewards as the effective ratio is rarely zero (i.e. all responses have the exact same reward). However, systematically understanding the role of prompt difficulty is valuable future work.
>
> > Weaknesses 2
>
> We agree that asynchronous pipelines can introduce additional off-policyness between the value model and policy, and we discuss the limitation of the synchronous setting in line 2061. We believe that PCL is reasonably robust to additional off-policyness for two reasons:
> - Policy updates are incremental, so $p_{\pi_t}(x)$ and $p_{\pi_{t-L}}(x)$ remain similar for small values of $L$. Typically, in asynchronous pipelines, L is between 2 to 4 which is reasonably small.
> - PCL does not require precise value estimates, and only needs a coarse relative ranking near a threshold. We include experiments using stochastic sampling based on a Beta distribution. The results are provided for the next point. When the Beta distribution has high $\alpha$ and $\beta$, the performance is similar to greedy sampling, showing that PCL is robust to noise.
>
> > Weaknesses 3
>
> During the development of PCL, we experiment with stochastic batch selection. Since our reward is binary, we sample from a Beta distribution whose support naturally lies in [0, 1]. We set $\alpha=\beta$ and ablate $\alpha$ and $\beta$ from 2 to 10, higher values indicate more focus on 0.5 difficulty prompts. From our experiments, greedy sampling and sampling based on Beta distribution perform similarly when $\alpha$ and $\beta$ are large. However, using smaller values of $\alpha$ and $\beta$, which spread the probability mass more broadly, actually degrades convergence, likely because it focuses less on 0.5 difficulty prompts. The results are detailed below for Llama3.2-3B-it on MATH dataset:
>
> | $\alpha$ | $\beta$ | MATH500 | Time to Convergence (hr) |
> |-------------|------------|---------------|--------|
> |       2       |      2      |  56.4         | 10.8 |
> | 5             | 5           |  56.2         | 15.9 |
> | 10           | 10         |  57.2         | 15.6 |
> | greedy   | greedy   |  **57.8**   | 14.3 |
>
> We deliberately chose greedy top-m selection because it is simple and robust to implement in large-scale RL pipelines, and already yields strong accuracy–time trade-offs compared to established baselines (GRPO, DS, SPEED, GRESO). We added a discussion of this result (line 2000 - 2020).
>
> In addition, we highlight that PCL progressively focuses on harder prompts during training, despite a fixed threshold of 0.5. We provide a detailed analysis in line 442. Although PCL maintains a fixed difficulty threshold of 0.5, as the policy improves, previously hard prompts would now appear intermediate, allowing PCL to continually shift toward more challenging examples.
>
> > Weaknesses 4
>
> We provide an ablation on the value model size in Appendix G, showing that a 4B value model performs similarly to a 8B model. This suggests that the size of the value model does not need to scale proportionally with the policy. While we have not tested larger policies due to resource constraints, we expect diminishing returns from larger value models.
>
> Moreover, for reasoning tasks, the prompt is typically much shorter than the response, which significantly reduces value model compute and memory compared to policy updates. As shown below, we include a detailed breakdown of the time per step when training Qwen3-8B-Base on MATH with three different sized value models (Qwen3-8B-Base, Qwen3-4B-Base, Qwen3-1.7B-Base). Value-related computation accounts for a small fraction of total time:
>
> | Policy | Value Model | Generation Time (s) | Value Computation Time (s) | Policy Update Time (s) | Value Model Update Time (s) |
> |-------------|------------|---------|--------|--------|--------|
> | Qwen3-8B-Base | Qwen3-8B-Base | 573 | **2.99** | 769 | **20.5** |
> | Qwen3-8B-Base | Qwen3-4B-Base | 567 | **2.11** | 763 | **12.4** |
> | Qwen3-8B-Base | Qwen3-1.7B-Base | 582 | **1.16** | 760 | **8.03** |
>
> Value computation time takes less than 1% of the generation time, and value model update time takes less than 3% of the time to update the policy.
>
> We again thank the reviewer for the insightful comments.

---

### Official Review · Reviewer_tnns · 2025-11-11

**Soundness:** 2
**Presentation:** 3
**Contribution:** 1
**Rating:** 2
**Confidence:** 5

**Summary:**

This paper re-uses classical active batch selection method (phrased as curriculum learning) in the post-training of large language models.
- Method: At each gradient update step, the method first samples a large pool of prompts, then chooses a subset from the pool as the training batch. The choice is based on signals from the value function.
- Experiments: The authors conduct the experiments using MATH and DeepScaleR, with models ranging from 1.7B to 8B. As in Table 1, the performance gain is relatively neutral.

**Strengths:**

- Presentation: The write-up is clear, and the descriptions of the experiments are detailed and easy-to-understand.
- Experiments: The attempt in understanding "optimal" batch size is very interesting (although there should be many artifacts in experimental settings that affect the optimality other than the factors proposed by the authors. It would be nice to further scale. One such example is [[Goyal et al., 2018](https://arxiv.org/pdf/1706.02677)]).

References:
[1] Goyal, Priya, Piotr Dollár, Ross Girshick, Pieter Noordhuis, Lukasz Wesolowski, Aapo Kyrola, Andrew Tulloch, Yangqing Jia, and Kaiming He. "Accurate, large minibatch sgd: Training imagenet in 1 hour." arXiv preprint arXiv:1706.02677 (2017).

**Weaknesses:**

- Lack of novelty: While the manuscript is well-written as a technical report, it offers limited methodological novelty or new insights to the field. The findings are well-known, and numerous prior works have explored similar directions. There appears to be little distinction between the proposed method and existing approaches, aside from the specific experimental setting (models/datasets). Consequently, this work seems better suited as a report or blog post, rather than a distinct contribution to academic research.
- Missing citations and comparisons: Related works should be put in the main body of the paper, instead of in the appendix. Also, please properly cite prior works on active batch selection and curriculum rl, e.g., [[Mindermann et al., 2020](https://arxiv.org/pdf/2206.07137)], [[Ash et al., 2019](https://arxiv.org/pdf/1906.03671)], [[Parker-Holder et al, 2022](https://arxiv.org/abs/2203.01302)], which all to some extent show it is better to learn from examples of medium difficulty. The proposed method appears to be a special case of many existing works, like [[Ye et al., 2024](https://arxiv.org/pdf/2411.00062)] but using greedy selection, or [[Muldrew et al., 2024](https://arxiv.org/pdf/2402.08114)] but using reinforce instead of dpo, or [[Kawaguchi et al, 2020](https://arxiv.org/pdf/1907.04371)] but using value as the selection criteria. A proper comparison with these works is necessary to accurately situate the paper within the fundamental literature and clarify the specific methodological contribution.
- Empirical performance: The empirical gains are marginal compared to existing baselines, in terms of both accuracy and computational efficiency. Further algorithmic exploration is needed to demonstrate significant performance improvements.

References:
[1] Kawaguchi, Kenji, and Haihao Lu. "Ordered sgd: A new stochastic optimization framework for empirical risk minimization." International Conference on Artificial Intelligence and Statistics. PMLR, 2020.
[2] Ye, Ziyu, Rishabh Agarwal, Tianqi Liu, Rishabh Joshi, Sarmishta Velury, Quoc V. Le, Qijun Tan, and Yuan Liu. "Scalable Reinforcement Post-Training Beyond Static Human Prompts: Evolving Alignment via Asymmetric Self-Play." ICML, 2024.
[3] Mindermann, Sören, Jan M. Brauner, Muhammed T. Razzak, Mrinank Sharma, Andreas Kirsch, Winnie Xu, Benedikt Höltgen et al. "Prioritized training on points that are learnable, worth learning, and not yet learnt." In International Conference on Machine Learning, pp. 15630-15649. PMLR, 2022.
[4] Ash, Jordan T., Chicheng Zhang, Akshay Krishnamurthy, John Langford, and Alekh Agarwal. "Deep batch active learning by diverse, uncertain gradient lower bounds." arXiv preprint arXiv:1906.03671 (2019).
[5] Parker-Holder, Jack, Minqi Jiang, Michael Dennis, Mikayel Samvelyan, Jakob Foerster, Edward Grefenstette, and Tim Rocktäschel. "Evolving curricula with regret-based environment design." In International Conference on Machine Learning, pp. 17473-17498. PMLR, 2022.

**Questions:**

I encourage the authors to consider more fundamental algorithmic improvements, instead of simply re-using old ideas as low-hanging fruit, which can be misleading. For example, can we do better than top-m subset selection? One related line of research is stochastic batch acquisition [[Kirsch et al., 2024](https://openreview.net/pdf/abd7cf88617c72ff4ec92264fb8a7919ef9ffee3.pdf)]. It would be helpful if the authors could provide a thorough discussion on the limitations of current work and include new experiments with better algorithms that bring something new to the academia. I am happy to raise my score if these concerns are sufficiently addressed.

References:
[1] Kirsch, Andreas, Sebastian Farquhar, Parmida Atighehchian, Andrew Jesson, Frederic Branchaud-Charron, and Yarin Gal. "Stochastic batch acquisition: A simple baseline for deep active learning." arXiv preprint arXiv:2106.12059 (2021).

---

> ### Author Response · Authors · 2025-11-22
>
> We thank the reviewer for the thoughtful feedback.
>
> > Experiments: The attempt in understanding "optimal" batch size is very interesting (although there should be many artifacts in experimental settings that affect the optimality other than the factors proposed by the authors. It would be nice to further scale. One such example is [1]).
>
> We appreciate the reviewer’s observation regarding potential artifacts influencing the optimal batch size. Extending this line of investigation is a valuable direction for future work.
>
> > Lack of novelty: While the manuscript is well-written as a technical report, it offers limited methodological novelty or new insights to the field...
>
> We respectfully disagree that our work is merely a re-use of classical active batch selection or curriculum learning without new insights. Our contributions are specific to **on-policy RL for LLM reasoning**, and go beyond what prior curriculum / active-learning papers have studied.
>
> Concretely, our contributions are:
> - **Systematic characterization of optimal batch size in RL for LLM post-training.** While large-batch phenomena have been studied in supervised settings [1], prior work has not studied the trade-offs between generation time and batch size for on-policy RL, nor identified a robust sweet spot in batch size in this setting. Our experiments show that:
>   - Generation time initially scales sublinearly, then linearly in batch size, due to the interplay between longest-sequence and compute utilization.
>   - There is a stable “sweet spot” batch size ($\approx$8K in our setting) at the transition point from sublinear to linear scaling that maximizes convergence speed, and this sweet spot is largely independent of how the batch is factorized into ($m$ prompts, $n$ generations) across models, datasets, hardware, and inference engines (line 198).
>   - To our knowledge, this systematic characterization under realistic RL pipelines for LLMs is new and practically important: **practitioners rarely know how to optimally choose batch size given a fixed time budget**.
> - **A new, theory-backed definition of the “medium difficulty” sweet spot for RL for LLMs.** Prior curriculum and active learning works have observed that “uncertain” examples are helpful. Our work:
>   - Derives the connection between the gradient norm and the prompt difficulty $p_{\pi}(x)$, showing that the gradient magnitude has the highest upper bound around with $p_{\pi}(x)=0.5$ for our RL objective (Appendix D).
>   - Empirically validates across multiple batch decompositions that $p_{\pi}(x)=0.5$ yields both higher effective ratios and higher test accuracy, and that blindly increasing the number of generations per prompt can hurt due to lost prompt diversity. **This unifies and sharpens the “medium difficulty” intuition in RL setting for LLM reasoning, where the interaction between $p_{\pi}(x)$, $n$, $m$ and convergence is non-trivial.**
> - **Prompt Curriculum Learning (PCL): a new value-based curriculum for LLM post-training that is compute efficient at scale.**
>   - The algorithm learns a value model over prompts online to approximate $p_{\pi}(x)$, using only the prompt as input, and updates it from the same on-policy rollouts used for RL without extra generations.
>   - At each step, PCL samples $km$ prompts, scores them with a single value-model forward pass, and greedily selects $m$ prompts closest to $\tau=0.5$ which eliminate the need for expensive looped sampling in Dynamic Sampling and SPEED as the responses are extremely long for reasoning tasks.
>   - This yields $12.1\times$ and $16.9\times$ faster difficulty estimation than rollout-based filtering on MATH and DeepScaleR, respectively. **Existing approaches either rely heavily on expensive rollouts (DS, SPEED) or depend on off-policy dictionaries that are inaccurate for large datasets (GRESO).**
> - **Large-scale empirical study (100K A100 GPU hours) across multiple model families and datasets.** Beyond an algorithm, we provide a carefully controlled, large-scale empirical study combining: multiple batch sizes and decompositions ($m$, $n$), multiple models (Qwen3 1.7/4/8B, Llama3.2-3B-it), and multiple math benchmarks (MATH500, Olympiad, Minerva, AMC23, AIME24/25). This scale and systematic exploration is, to our knowledge, unprecedented for analyzing the batch size trade-offs for RLVR-style LLM training, and yields actionable guidance for practitioners that goes beyond an anecdotal or blog-level study.
>
> We clarified these contributions in the introduction more explicitly (line 45 - 47).

---

> ### Author Response · Authors · 2025-11-22
>
> > Missing citations and comparisons: Related works should be put in the main body of the paper, instead of in the appendix. Also, please properly cite prior works on active batch selection and curriculum rl...
>
> We appreciate these pointers and agree that a more prominent and fine-grained positioning will improve the paper. We moved the Related Work into the main body as the page limit would be increased for camera ready, and explicitly discussed the suggested works. Here we summarize the main distinctions:
> - Mindermann et al. 2020 [2] prioritizes training on points that are learnable, worth learning, and not yet learned using a loss-based criterion in supervised learning. Their notion of “worth learning” is tied to minimizing the loss on a holdout set.
>   - Difference: We work in on-policy RL for long CoT LLMs with binary verifiable rewards, where the gradient structure is fundamentally different. Our setting is much more complicated with more hyperparameters (batch size, $m$, $n$, etc.) and frameworking involving online generations from the current policy without a holdout set.
> - Ash et al. 2019 [3] proposes BADGE which uses gradient embeddings to select diverse and uncertain examples in supervised classification.
>   - Difference: BADGE operates with access to per-sample gradients and labels in a static pool. By contrast, our setting lacks ground-truth rollouts, and each rollout contains thousands of tokens which are thousands of classification tasks. We only observe binary rewards at the end of long trajectories and must deal with high-variance policy gradients, which leads to a different analysis and algorithmic design (value model with medium difficulty prompts).
> - Parker-Holder et al. 2022 [4] proposes ACCEL which is a regret-based environment design that evolves environment parameters to shape a curriculum in RL.
>   - Difference: ACCEL aims to propose the simplest levels that the agent cannot currently solve and modify the environment distribution in the setting of UPOMDP. We instead define a data curriculum over prompts in a fixed distribution and study its interaction with batch configuration and rollout cost in LLM post-training. In addition, under the binary reward setting, the minimax-regret Nash equilibrium of ACCEL corresponds to a teacher that concentrates on the hardest solvable environments for that policy.  This predicted equilibrium does not match our empirical finding that the optimal difficulty concentrates around 0.5 difficulty prompts. This discrepancy demonstrates that our observations capture a previously unreported phenomenon and that our contribution is novel rather than a reuse of ACCEL’s insight.
> - Ye et al. 2024 [5] proposes asymmetric self-play that evolves tasks and policies via a sophisticated two-player scheme. This involves task generation / self-play and requires substantial additional rollout cost.
>   - Difference: Ye et al. focus on evolving the task distribution via self-play while we focus on re-weighting an existing verifiable dataset through a learned value model. Their setting is not centered on the convergence analysis while we characterize how to use limited rollouts efficiently in large-scale math RL. In addition, the paper uses the same minimax-regret formulation as ACCEL. As with ACCEL, this equilibrium predicts concentration on the hardest solvable tasks which contradicts our empirical observation that optimal performance arises near 0.5 difficulty. This further underscores that our findings reveal a new and previously uncharacterized phenomenon.
> - Muldrew et al. 2024 [6] proposes active preference learning with DPO, selecting pairs for preference models.
>   - Difference: They study preference optimization with DPO-style objectives, using uncertainty in preference labels as a selection signal for active learning. We study online RL with verifiable rewards and focus on improving convergence. The settings and focuses of the two papers are completely different.
> - Kawaguchi & Lu 2020 [7] analyze the effect of sample ordering on ERM in standard supervised learning.
>   - Difference: Their Ordered SGD framework assumes access to a static dataset with inexpensive per-sample gradients, making sample ordering a lightweight modification of classical stochastic optimization. In contrast, our setting involves expensive on-policy rollouts, long-context sequence generation, and costly gradient computation in large-scale LLM post-training. As a result, the assumptions, computational constraints, and algorithmic considerations in our work differ substantially from those in the Ordered SGD setting.
>
> We added a dedicated “Curriculum and Active Learning” subsection in related work (line 497 - 509) that covers [2 - 7] and clarifies that PCL is not simply a re-branding, but adapts and extends these ideas to the specific constraints and structure of RLVR-style LLM post-training.

---

> ### Author Response · Authors · 2025-11-22
>
> > Empirical performance: The empirical gains are marginal compared to existing baselines, in terms of both accuracy and computational efficiency. Further algorithmic exploration is needed to demonstrate significant performance improvements.
>
> We agree that raw accuracy differences alone can look small in some settings, and we appreciate the opportunity to clarify the joint accuracy–time gains, which we view as the main empirical contribution.
>
> **Faster convergence at a similar or better accuracy.** As shown in Table 1 and Appendix F, PCL consistently achieves either (i) the highest final accuracy or (ii) substantially reduced wall-clock time at comparable accuracy. On MATH, PCL achieves the best test accuracy across all model sizes and families. On DeepScaleR, PCL either achieves the best average accuracy or converges significantly faster. We summarizes the results on DeepScaleR below from Appendix F, comparing PCL with the best or the second best method for each model in terms of test accuracy:
> - Qwen3-8B-Base: PCL is the top-performing method and converges in 41.8 hours, compared to 69.5 hours for DS, which also achieves lower average accuracy. PCL converges **39.8%** faster.
> - Qwen3-4B-Base: PCL achieves 45.7 test accuracy and converges in 32.8 hours, compared to 45.8 accuracy and 40.1 hours for DS. PCL therefore converges **18.2%** faster while maintaining similar performance.
> - Qwen3-1.7B-Base: PCL is again the best-performing method and converges in 23.3 hours, whereas Pre-filter requires 44.2 hours and yields lower average accuracy. PCL converges **47.3%** faster.
> - Llama3.2-3B-it: PCL reaches 26.5 accuracy in 28.7 hours, compared to 26.7 accuracy in 40.6 hours for DS, achieving similar accuracy with **29.3%** faster convergence.
>
> The consistent **1.3 to 2$\times$** reductions in convergence time are practically impactful in real RL pipelines where rollouts are the dominant cost. To ensure this contribution is clearer, we updated the paper to emphasize the efficiency perspective (line 350 - 352, 1892 - 1910).
>
> > I encourage the authors to consider more fundamental algorithmic improvements, instead of simply re-using old ideas as low-hanging fruit, which can be misleading. For example, can we do better than top-m subset selection? One related line of research is stochastic batch acquisition [8]. It would be helpful if the authors could provide a thorough discussion on the limitations of current work and include new experiments with better algorithms that bring something new to the academia.
>
> We appreciate this suggestion and agree it is a promising direction. The PCL framework is compatible with stochastic batch acquisition in the sense of Kirsch et al. (2021): given scores $s(x) = |V(x) − \tau|$, one could sample the batch from a distribution favoring lower $s(x)$ instead of taking the deterministic bottom m prompts.
>
> During the development of PCL, we experiment with stochastic batch selection. Since our reward is binary, we sample from a Beta distribution whose support naturally lies in [0, 1]. We set $\alpha=\beta$ and ablate $\alpha$ and $\beta$ from 2 to 10, higher values indicate more focus on 0.5 difficulty prompts. From our experiments, greedy sampling and sampling based on Beta distribution perform similarly when $\alpha$ and $\beta$ are large. However, using smaller values of $\alpha$ and $\beta$, which spread the probability mass more broadly, actually degrades convergence, likely because it focuses less on 0.5 difficulty prompts. The results are detailed below for Llama3.2-3B-it on MATH dataset:
>
> | $\alpha$ | $\beta$ | MATH500 | Time to Convergence (hr) |
> |-------------|------------|---------------|--------|
> |       2       |      2      |  56.4         | 10.8 |
> | 5             | 5           |  56.2         | 15.9 |
> | 10           | 10         |  57.2         | 15.6 |
> | greedy   | greedy   |  **57.8**   | 14.3 |
>
> We deliberately chose greedy top-m selection because it is simple and robust to implement in large-scale RL pipelines, and already yields strong accuracy–time trade-offs compared to established baselines (GRPO, DS, SPEED, GRESO). We added a discussion of this result (line 1998 - 2020), and explicitly acknowledge that stochastic variants remain a promising future direction. We also note that the Limitation section is provided in Appendix J and already highlights several axes not yet explored, such as off-policy and asynchronous pipelines.
>
> We hope this clarifies that our work goes beyond reusing old ideas, provides new insights that are specific to RL-based LLM post-training, and offers practical guidance for large-scale training pipelines.

---

> ### Author Response · Authors · 2025-11-22
>
> References:
>
> [1] Goyal, Priya, Piotr Dollár, Ross Girshick, Pieter Noordhuis, Lukasz Wesolowski, Aapo Kyrola, Andrew Tulloch, Yangqing Jia, and Kaiming He. "Accurate, large minibatch sgd: Training imagenet in 1 hour." arXiv preprint arXiv:1706.02677 (2017).
>
> [2] Mindermann, Sören, Jan M. Brauner, Muhammed T. Razzak, Mrinank Sharma, Andreas Kirsch, Winnie Xu, Benedikt Höltgen et al. "Prioritized training on points that are learnable, worth learning, and not yet learnt." In International Conference on Machine Learning, pp. 15630-15649. PMLR, 2022.
>
> [3] Ash, Jordan T., Chicheng Zhang, Akshay Krishnamurthy, John Langford, and Alekh Agarwal. "Deep batch active learning by diverse, uncertain gradient lower bounds." arXiv preprint arXiv:1906.03671 (2019).
>
> [4] Parker-Holder, Jack, Minqi Jiang, Michael Dennis, Mikayel Samvelyan, Jakob Foerster, Edward Grefenstette, and Tim Rocktäschel. "Evolving curricula with regret-based environment design." In International Conference on Machine Learning, pp. 17473-17498. PMLR, 2022.
>
> [5] Ye, Ziyu, Rishabh Agarwal, Tianqi Liu, Rishabh Joshi, Sarmishta Velury, Quoc V. Le, Qijun Tan, and Yuan Liu. "Scalable Reinforcement Post-Training Beyond Static Human Prompts: Evolving Alignment via Asymmetric Self-Play." ICML, 2024.
>
> [6] Muldrew, William, Peter Hayes, Mingtian Zhang, and David Barber. "Active preference learning for large language models." arXiv preprint arXiv:2402.08114 (2024).
>
> [7] Kawaguchi, Kenji, and Haihao Lu. "Ordered sgd: A new stochastic optimization framework for empirical risk minimization." International Conference on Artificial Intelligence and Statistics. PMLR, 2020.
>
> [8] Kirsch, Andreas, Sebastian Farquhar, Parmida Atighehchian, Andrew Jesson, Frederic Branchaud-Charron, and Yarin Gal. "Stochastic batch acquisition: A simple baseline for deep active learning." arXiv preprint arXiv:2106.12059 (2021).

---

### Comment · Area_Chair_H35z · 2025-11-25
**Reminder of PDF Update and Discussions**

Hi Author,

Remember to update the manuscript pdf for the reviewers and ACs to examine. This could be crucial for the final evaluation in meta-review. The deadline for the author-reviewer discussions and pdf update is Dec. 3rd.

Best,
AC

Hi Reviewers,

Please carefully read the rebuttal from the author and update the review and score accordingly if the concerns are well addressed. Please note that the Author-Reviewer discussion will end after one week.

Best,
AC

---

### Author Response · Authors · 2025-11-26

We thank all reviewers for their detailed and constructive feedback. We have revised the draft to incorporate the following changes:

- Made our contributions more explicit (lines 45–47).
- Placed stronger emphasis on the efficiency perspective of PCL (lines 350–352, 1892–1910).
- Moved the related work section into the main paper and added a new subsection on curriculum and active learning (lines 497–509).
- Cited additional related work, including MoPPS and MPTS (lines 487–489, 524–526).
- Added discussion and experiments on stochastic batch selection (lines 529–534, 1998–2020).
- Included a detailed training time breakdown (lines 2025–2040).
- Added a discussion of limited task diversity in the limitations section (lines 2077–2084).

Thank you again for your time and effort in reviewing our work.

---

### Meta-Review · Area_Chair_H7if · 2026-01-09

**Summary:**

The work evaluates 1) the effect of batch size and number of generations 2) and a curriculum learning approach which keeps the pass rate. around 50%. The curriculum learning approach is based on value function.

**Reviewer Concerns:**

Reviewers’ main concerns are that the method appears incremental, largely reusing known curriculum/active selection ideas and that the empirical accuracy gains over strong baselines are often small, with evaluation concentrated mostly on math tasks. They also flag risks: reliance on a possibly lagging value model, potential loss of diversity or distributional bias from focusing on ~50% difficulty and the overhead of maintaining a second model.

**Reviewer Scores:**

All reviewers are positive about the paper, except one which recommends a sharp rejection (score 2). It is unlikely that such reviewer would fully flip its opinion to accepting the paper, as the comments are structural.

---

### Decision · Program_Chairs · 2026-01-26

Accept (Poster)